# Optimized model architectures for deep learning on genomic data
Hüseyin Anil Gündüz[1,2], René Mreches[3,4], Julia Moosbauer [1,2], Gary Robertson[3,4], Xiao-Yin To [1,2,3,4], Eric A. Franzosa [5], Curtis Huttenhower [5], Mina Rezaei [1,2], Alice C. McHardy [3,4,6], Bernd Bischl[1,2], Philipp C. Münch [3,4,5,6,7] ✉ & Martin Binder [1,2,7] ✉

The success of deep learning in various applications depends on task-specific architecture design choices, including the types, hyperparameters, and number of layers. In computational biology, there is no consensus on the optimal architecture design, and decisions are often made using insights from more well-established fields such as computer vision. These may not consider the domain-specific characteristics of genome sequences, potentially limiting performance. Here, we present GenomeNet-Architect, a neural architecture design framework that automatically optimizes deep learning models for genome sequence data. It optimizes the overall layout of the architecture, with a search space specifically designed for genomics. Additionally, it optimizes hyperparameters of individual layers and the model training procedure. On a viral classification task, GenomeNet-Architect reduced the read-level misclassification rate by 19%, with 67% faster inference and 83% fewer parameters, and achieved similar contig-level accuracy with ~100 times fewer parameters compared to the best-performing deep learning baselines.

Deep learning (DL) techniques have been shown to achieve exceptional performance on a wide range of machine learning (ML) tasks, especially when large training sets are available[1]. These techniques have been applied to a variety of challenges in bioinformatics[2–4]. For different ML problems and data modalities, different neural architectures have emerged that perform well in their respective domains, such as convolutional neural networks (CNN) for images or recurrent neural networks (RNN) for text. Architectural design choices are often made based on the experience of researchers and trial and error[5–13]. However, the optimal design and arrangement of these layers are highly domain-specific, problem-dependent, and computationally expensive to evaluate. Besides expert-driven design, it has therefore become increasingly popular to apply systematic approaches to finding neural network configurations, such as automated neural architecture search (NAS)[14]. The number of possible configurations of even small neural networks is very large, as the number of decisions to be made grows exponentially, and most practical NAS algorithms therefore impose various constraints on the search space.

To efficiently perform NAS for ML tasks in genomics, it is essential to identify DL network architecture designs for genomic sequence analysis that are widely recognized in the literature. These designs often start with one or several convolutional layers, followed by a global pooling layer, and conclude with a series of fully connected layers[6,7,9,11]. Recurrent layers offer an alternative to convolutional or global pooling layers. Their ability to propagate information across sequences allows recurrent layers to effectively summarize data, comparable to pooling layers. While numerous works in genomics use RNN layers[15–19], one example is *Seeker*[8], an RNN-based model that employs an LSTM layer for bacteriophage detection. Furthermore, by stacking them sequentially, integrating convolutional and recurrent layers enhances model capability. For instance, the model developed by Wang et al.[20] demonstrates this approach by placing an RNN on top of a convolutional layer, followed by two fully connected layers. A similar configuration is utilized in the DanQ model[21], showcasing the effectiveness of combining recurrent and convolutional layers.

One way to approach NAS is to consider it as a hyperparameter optimization (HPO) problem. Hyperparameters (HPs) are configuration settings that determine how an ML model works. In the context of DL, typical HPs are the choice of the gradient descent algorithm and its learning rate. However, the choice of neural network layers and their configuration can also be considered as HPs. NAS is then equivalent to optimizing HPs that define different

[1]Department of Statistics, LMU Munich, Munich, Germany. [2]Munich Center for Machine Learning, Munich, Germany. [3]Department for Computational Biology of Infection Research, Helmholtz Center for Infection Research, 38124 Braunschweig, Germany. [4]Braunschweig Integrated Centre of Systems Biology (BRICS), Technische Universität Braunschweig, Braunschweig, Germany. [5]Department of Biostatistics, Harvard School of Public Health, Boston, MA, USA. [6]German Centre for Infection Research (DZIF), partner site Hannover Braunschweig, Braunschweig, Germany. [7]These authors jointly supervised this work: Philipp C. Münch, Martin Binder. ✉e-mail: philipp.muench@helmholtz-hzi.de; martin.binder@stat.uni-muenchen.de

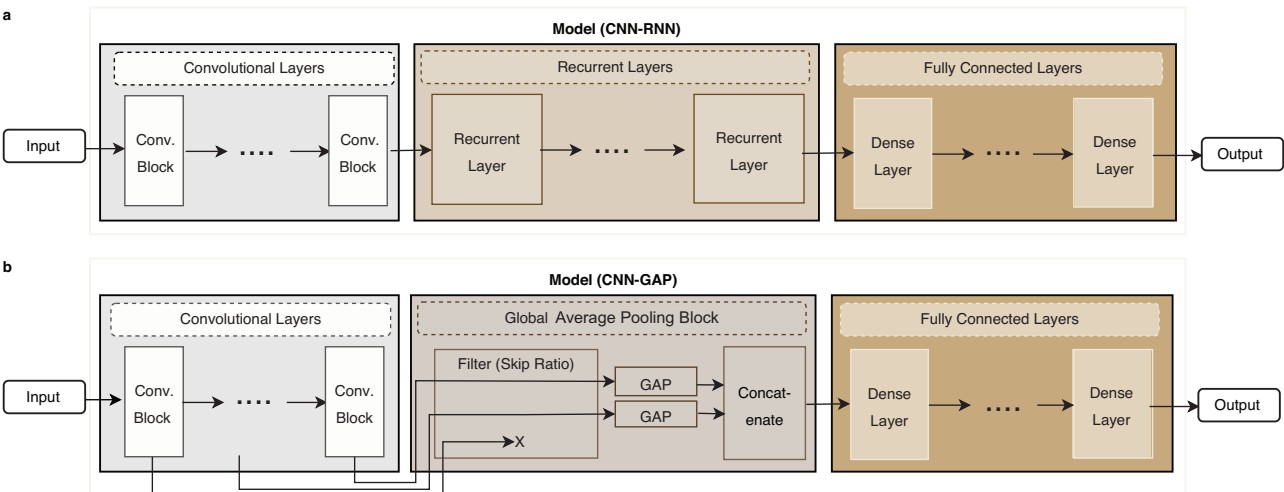

**Fig. 1 | The network layout optimized by GenomeNet-Architect consists of three stages: (i) a stage of stacked convolutional layers, (ii) global average pooling (in the CNN-GAP model) or a stack of recurrent layers (in the CNN-RNN model), and (iii) a fully connected stage. a** The CNN-RNN model feeds the output of the last convolutional layer into a block of recurrent layers. The output of the last recurrent layer is then flattened and fed into a fully connected neural network. **b** The CNN- GAP model groups the convolutional layers into convolutional blocks. While the output of some of these blocks is skipped (controlled by the "skip ratio" hyper-parameter), the network performs global average pooling (GAP) on the remaining blocks and concatenates the result. This is then fed into the fully connected neural network.

architectures. A popular class of optimization algorithms used for HPO is model based optimization (MBO), also called Bayesian optimization[22]. It iteratively evaluates HP configurations and selects new configurations to try based on knowledge of which configurations have worked well in the past. This is done by fitting a regression model, the so-called surrogate model, to the observed performance values. New evaluations are made by considering the "exploration-exploitation tradeoff": new configurations should be tried if their expected performance is high (exploitation), or if the model's uncertainty about their performance is high (exploration). MBO-based methods such as BANANAS[23] have been shown to outperform methods based on other optimization paradigms such as ENAS[24] (which uses reinforcement learning) or DARTS[25] (which uses gradient descent).

The quality of the configurations evaluated by MBO increases gradually as the optimization progresses. The first configurations evaluated, which constitute the initial design, are randomly sampled from the search space without using any prior knowledge. By anticipating that early configurations are unlikely to perform as well as later ones, and by devoting fewer resources to their evaluation, it is possible to reduce the cost of the overall optimization process. Algorithms that speed up optimization by using cheaper approximations of the target objective are called multi-fidelity (MF) optimization algorithms. A simple way to approximate the performance of a DL model is to stop training the model after a certain amount of time, even though the model performance has not fully converged[26].

While there are libraries that perform NAS on genome datasets[27], we are not aware of any methods that use efficient multi-fidelity or MBO methods specifically for genome datasets. MBO has been used in the past to tune specifically designed genomic DL models[28], but only to optimize specific HPs, not as a general NAS framework. No general-purpose MBO-based NAS framework provides a search space specifically modified to fit genome sequence data; in fact, many focus on 2D image data instead.

In this work, we present GenomeNet-Architect, which optimizes DL network architectures by repeatedly constructing new network configurations, training networks based on these configurations on a given dataset, and evaluating the performance of the resulting models by predicting on held-out test data. It uses MBO as an efficient black-box optimization method, combined with a multi-fidelity approach that increases model training time after some initial optimization progress has been made. Unlike other general-purpose NAS frameworks, GenomeNet-Architect uses a search space specifically for genome data. It is made up of neural

architectures and HP setups that build on top of and generalize various architectures for the genome data that have been successfully applied in the past. This approach allows us to efficiently explore a large space of possible network architectures and identify those that perform well for genome-related tasks, creating architectures that outperform expert-guided architectures. Our method can be used for a variety of DL tasks on genome sequence data, such as genome-level, loci-level, or nucleotide-level classification and regression.

## Results
### GenomeNet-Architect uses an efficient global optimization method
GenomeNet-Architect provides a predefined search space of hyperparameters (HPs) that are used to construct different network architectures. It needs to be given a specific ML task on genome sequence data. In our framework, we use model-based optimization (MBO)[22] to jointly tune the network layout and HPs, and generate a specific architecture that works well on the given task.

The result of the optimization process itself is a specific HP configuration that works well for the given task. The resulting architecture can be trained and evaluated on the given data, as well as used to make predictions on new data. However, the resulting architecture can also be used for other tasks that are similar to the task for which it was optimized. It is therefore possible to perform a single optimization run to solve multiple genome sequence DL tasks.

### GenomeNet-Architect uses a search space that covers the most common layer types and hyperparameter settings
The search space of GenomeNet-Architect is based on our literature analysis of successful architectures developed for genome data, such as DeepVirFinder[6], ViraMiner[7], Seeker[8], CHEER[9], Fiannaca (CNN model)[10], PPR-Meta[11], and an adapted version of RC-ResNet-18[12]. A common type of architecture consists of convolutional layers followed by global pooling and fully connected layers[6,7,9,11]. An alternative to pooling, which also aggregates information across the entire sequence, is the use of recurrent (RNN) layers.

Inspired by these common patterns observed in many networks successfully applied to genome data, we build a template for an architecture consisting of three stages (Fig. 1): (i) a stage of stacked convolutional layers

**Table 1 | Space of hyperparameters that affect the training and final layout of the model, along with the ranges over which they are optimized**

| Hyperparameter | Type | Range | Log-Search Space | Component |
|---|---|---|---|---|
| Learning Rate ($l_r$) | Float | $[10^{-6}, 10^{-2}]$ | ✓ | General |
| Reverse-Complement as Additional Input | Boolean | {*True, False*} | | |
| Optimizer | Categ | {*Adam, Adagrad, Rmsprop, Sgd*} | | |
| Model Type | Categ | {*GAP, RNN*} | | |
| Number of Convolutional Layers ($n_c$) | Integer | $[1,20]$ | | Convolutional Layers |
| Number of Convolutional Blocks ($n_{cb}$) | Integer | $[1,10]$ | | |
| First Layer Kernel Size ($k_0$) | Float | $[2^4,2^{11}]$ | ✓ | |
| Last Layer Kernel Size ($k_{end}$) | Float | $[2^4, 2^{11}]$ | ✓ | |
| First Layer Number of Filters ($f_0$) | Float | $[2^1,2^6]$ | ✓ | |
| Last Layer Number of Filters ($f_{end}$) | Float | $[2^1,2^6]$ | ✓ | |
| Last Layer Dilation Factor ($d_{end}$) | Float | $[2^0,2^4]$ for $L = 150$ or $250$, $[2^0,2^7]$ for $L = 10000$ | ✓ | |
| Total Max-Pooling ($p_{end}$) | Float | $[2^0,2^4]$ for $L = 150$ or $250$, $[2^0,2^7]$ for $L = 10000$ | ✓ | |
| Momentum of Batch-Normalization | Float | $[0,0.99]$ | | |
| Leaky-ReLU Alpha Value | Float | $[0,1]$ | | |
| Residual Block (*res_block*) | Boolean | {*True, False*} | | |
| Number of Dense Layers | Integer | $[0,5]$ | | Fully Connected Layers |
| Units of Dense Layers | Float | $[2^4,2^{11}]$ | ✓ | |
| Dropout of Dense Layers | Float | $[0,0.99]$ | | |
| Activation of Dense Layers | Categ | {*ReLU, tanh, Sigmoid*} | | |
| Recurrent Layer Type | Categ | {*LSTM, GRU*} | | Recurrent Layers (CNN-RNN only) |
| Number of Recurrent Layers | Integer | $[1,3]$ | | |
| Uni-/Bidirectional Recurrent Layers | Boolean | {*True, False*} | | |
| Number of Recurrent Units | Float | $[2^4, 2^{11}]$ | ✓ | |
| Skip Ratio for Global Average Pooling ($r_s$) | Float | $[0,1]$ | | Global Average Pooling Block (CNN-GAP only) |

Where indicated in the "Log-Search Space" column, hyperparameters are optimized on a logarithmic scale. Hyperparameters are grouped by the "Component" they control, corresponding to the different components shown in Fig. 1.

operating on one-hot encoded input sequences, (ii) a stage for embedding the sequential output data of the convolutional layers into a vector representation, using either global average pooling (GAP) (in a setup that we call the CNN-GAP model) or a stack of recurrent layers (which we call the CNN-RNN model), and (iii) a fully connected neural network stage operating on the embedded values.

Some of the properties we search over include the network layout, such as the number and size of convolutional, dense, and recurrent layers. Other HPs that we searched over influence the training process, such as the optimizer, and the behavior of specific layers, such as the dropout rate, the activation functions, and the batch normalization constant (Table 1). By introducing multiple HPs that influence the final layout of the model, our framework covers many successful architectures from the literature, while also making it possible to find architectures that have not yet been implemented. While the provided search space is our recommendation, our method also supports defining a custom search space, e.g. allowing more layers, or including GMP instead of GAP.

Our HPs cover both the overall architecture of the network (e.g., number of convolutional layers) and the setup of individual layers (e.g., the CNN kernel size). Having different HPs for each layer individually would introduce HP dependencies, which would make the optimization problem more difficult. Therefore, we use a setup where only the first and last layers are directly parameterized. $f_0$ and $f_{end}$, for example, specify the number of filters of the first and last convolutional layer. The setup of the intermediate layers is interpolated based on the first and last layers (see Methods for more details).

**Model configurations are initially evaluated with shorter runtimes for more efficient search space exploration**

Several challenges arise when optimizing DL architectures on complex data modalities such as genomics. First, for complex tasks, the time required for a single model to converge to a solution makes it impractically slow to optimize over a large search space such as the one we have designed. A simple way to speed up model evaluation would be to limit the time for which each proposed model is trained, even if models do not converge within a given timeframe, because models that perform well early in model fitting will continue to perform well after more training epochs[26]. While this reduces the time spent on individual evaluations, the resulting models can only approximate the true performance of a given HP configuration. Smaller models (which have fewer parameters and therefore converge faster) may falsely appear to be superior to larger models that run slower, complete fewer epochs, and cannot converge in the given time limit. However, the models trained for only a short time are still informative about which parts of the search space are more likely to contain models that perform well. We can therefore use them in a "warm start" method that speeds up the optimization process. This works by only partially evaluating initial configuration proposals at first, and using the resulting data about which HPs tend to perform well for short evaluation times to help determine which configurations are later evaluated for longer training times[29].

GenomeNet-Architect first runs the MBO with a fixed, low setting for the model training time $t = t_1$. After a given number of optimization iterations, a new MBO run is started with a higher training time setting $t = t_2$, where the surrogate model contains the performance result data for $t = t_1$ as

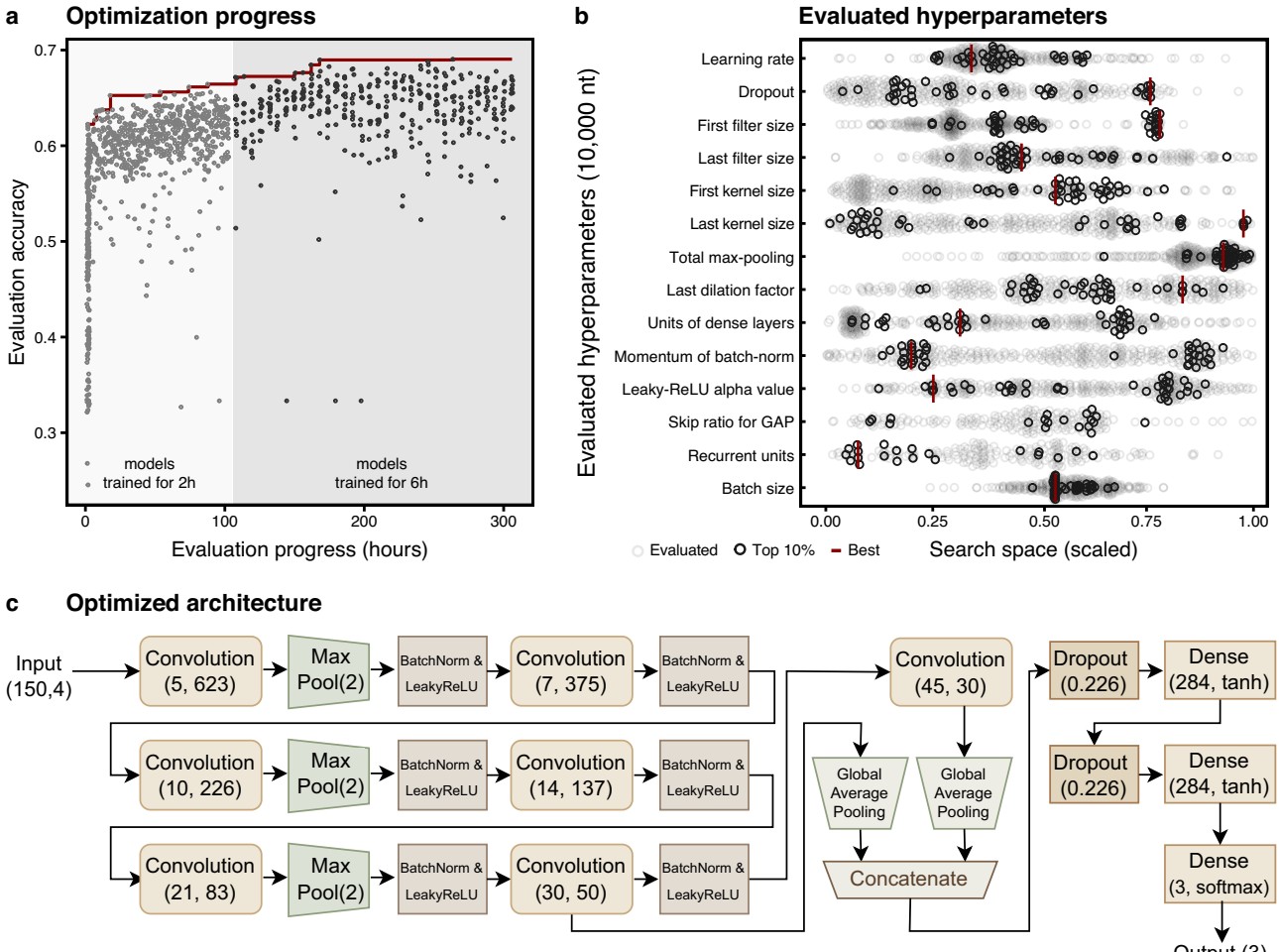

**Fig. 2 | Overview of the optimization procedure and results on the viral classification task.** We evaluated the performance of the models proposed by our optimization framework on a viral classification task and compared them to various baseline methods for sequence lengths of 150 and 10,000 nt to evaluate for application at the read and (large) contig level (Table 2). **a** The progress of hyperparameter optimization for 150 nt sequence length. Models are trained for 2 h and then 6 h at different stages of the optimization. A better set of hyperparameters is discovered as the optimization proceeds. **b** Evaluated values in the hyperparameter optimization for 6 h of training time and a sequence length of 10,000 nt. The search range (Table 1) of each hyperparameter is normalized in the plot. Dark circles indicate the top 10% of evaluated configurations clustered around favorable values. The best-selected configuration (vertical red lines) often lies within this cluster. **c** The model selected by the hyperparameter optimization stage for a sequence length of 150 is shown (CNN-GAP-6 h in Supplementary Table 1). The values in parentheses indicate kernel size and number of filters for convolutional layers, units and activation for dense layers, and both pool size and stride values for max-pooling layers. Since $n_{cb}$ is 7, there are seven convolutional layers with increasing kernel size and decreasing number of filters along the model. Since the GAP skip ratio is 72%, only the outputs of the last two convolutional layers are pooled and concatenated in the GAP block. This is followed by two dense hidden layers with tanh-activation.

a "warm start". The optimization procedure can be restarted several times with higher values of $t_n$ using all previous points evaluated at times $t_1, ..., t_{n-1}$ as warm start data. Our experiments started with $t_1 = 2$ h and then continued with $t_2 = 6$ h (Fig. 2a). Longer times were also tried, but did not lead to sufficient improvement to justify the additional resources required (Supplementary Note 1). Models evaluated after increasing the training time have higher performance than randomly sampled models at the beginning, showing that the information learned from models trained for a short time is useful for building models trained for a longer time.

**GenomeNet-Architect makes use of parallel resources**
GenomeNet-Architect parallelizes the HP tuning process across multiple GPUs to reduce the overall optimization time. There are a variety of multipoint proposal techniques that allow MBO to evaluate several different HP configurations simultaneously[30]. A particularly straightforward method is to use the UCB (upper confidence bound) infill criterion[31]: Given a parameter $\lambda$, it uses the mean prediction of the surrogate model and adds $\lambda$ times the model's uncertainty, thereby giving an optimistic bias to regions that have

high uncertainty and therefore potential for improvement. By sampling multiple instances of the $\lambda$-parameter from an exponential distribution[32], effectively making different tradeoffs between exploration and exploitation, one can generate different point propositions to be evaluated simultaneously. It is used by our method because, despite its simplicity, it is one of the best-performing MBO parallelization methods[30].

In addition to parallelizing individual MBO runs, we also identify setups that are likely to lead to different optimal configurations, and whose optimization can therefore be run independently and in parallel: Which HP settings are optimal, such as specific kernel sizes or number of filters, may vary for different sequence lengths. Similarly, optimal values may differ for CNN-GAP and CNN-RNN, and they may also change depending on whether residual blocks are used. Therefore, the optimization proceeds in a fully crossed design of these choices: The length of the training minisequence (in our experiments we investigated both 150 nt, 250 nt, and 10,000 nt), the architecture (CNN-GAP or CNN-RNN), and whether residual connections are used. These optimization runs are independent of each other and can be run in parallel.

**Fig. 3 | Predictive performance and characteristics of models found by GenomeNet-Architect and various baselines on the viral classification task.** The inference time to classify 10,000 one hot encoded samples on a GPU and the class-balanced accuracy are shown. The size of the circles indicates the number of model parameters. We have not included Seeker[8] in both graphs and PPR-Meta[11] for long sequences because their performance was too low. **a** At the read-level (150 nt), the best-performing model selected by GenomeNet-Architect (CNN-GAP-6h) reduces the read-level misclassification rate by 19% relative to the best-performing deep learning baseline - Fiannaca[10], despite having 83% fewer parameters and 67% faster inference time. **b** At the contig-level (10,000 nt), models found by GenomeNet-Architect perform on par or better than the best-performing baseline, although all these models perform very well in terms of accuracy. However, much faster (CNN-GAP-2h) and much smaller (CNN-RNN-6h) models are found, all with balanced accuracy close to the best baseline ( ~ 98.6%).

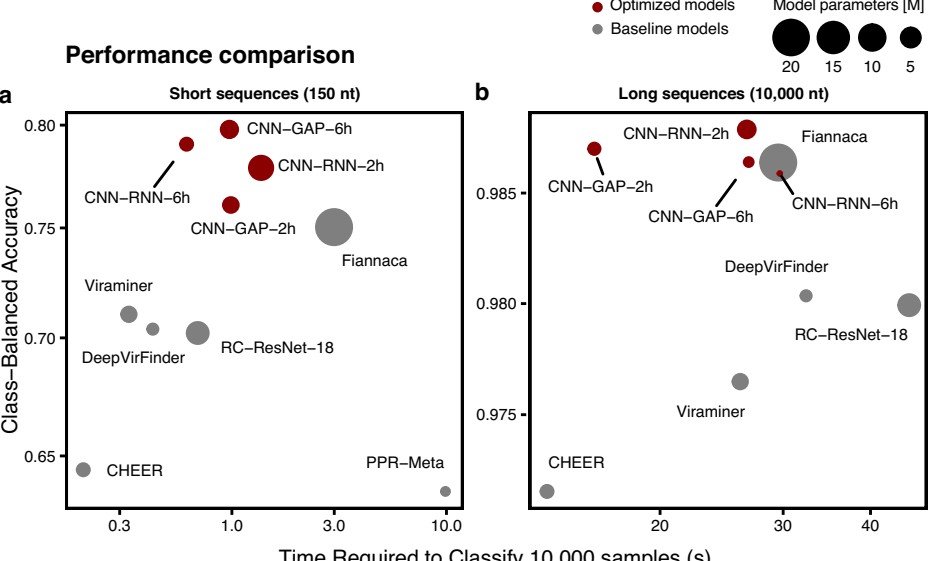

## GenomeNet-Architect finds models that outperform expert-designed baseline models in the viral identification task

GenomeNet-Architect demonstrated superior performance on the virus classification task compared to other deep learning (DL) and non-deep learning methods that we selected as baselines, effectively distinguishing between sequences originating from bacterial chromosomes, prokaryotic viruses (referred to as bacteriophages) and eukaryotic viruses (referred to as viral non-phage DNA). We have tested the effectiveness of GenomeNet-Architect against baselines for classification at the read-level (150 nt long sequences) and at the contig-level (10,000 nt) separately. At the read level, GenomeNet-Architect reduces the class-balanced misclassification rate, i.e., the misclassification rate averaged over all classes, by 19%, while having 83% fewer parameters and achieving 67% faster inference time compared to the best DL baseline (Fiannaca[10]) and outperforms k-mer-based and alignment-based approaches for sequence classification (Fig. 3a, Supplementary Table 2). At the contig-level, the best model found by our method achieves a class-balanced misclassification rate of 1.21%, outperforming the best baseline (1.36%) while being 82% smaller. GenomeNet-Architect also finds a model that performs comparably to the baseline (1.41%) while having a factor of 117 times fewer parameters (Fig. 3b, Supplementary Table 1). For a fair comparison, we trained and validated all DL baseline models on the same dataset and dataset splits. Additionally, we standardized the configuration by adapting the output layer of each model and employing multi-class cross-entropy as the loss function, aligning with our models to facilitate three-class classification. This approach allows for a direct comparison of algorithmic improvements.

We show the best configuration found for this classification task using 10,000 nt sequences (red lines), as well as the top 10% configurations (solid circles) in front of all evaluated configurations (transparent circles) (Fig. 2b) and a diagram visualizing the optimized architecture for the classification task using 150 nt (Fig. 2c). We also analyzed the performance of the optimized model stratified by the degree of genomic differences to the training data. Our findings show that it consistently outperforms the Fiannaca baseline, demonstrating that GenomeNet-Architect's performance does not come from overfitting on sequences that occur in the training dataset (Supplementary Fig. 1). The superior performance persists even when reads are simulated with the Illumina read error profile, underscoring the tool's effectiveness across diverse genomic sequencing challenges (Supplementary Fig. 2).

## GenomeNet-Architect identifies models that outperform expert-designed baselines in the pathogenicity detection task

To further validate the versatility of GenomeNet-Architect, we extended our experiments to a second task: pathogen detection in bacteria, specifically to distinguish between pathogenic and non-pathogenic sequences in human hosts. We aligned our evaluation with the baseline values reported in the study of Bartoszewicz et al.[13], utilizing the same search space for the viral classification task and hyperparameter optimization stage, where models are optimized for 2 h. We also fine-tuned the pre-trained DNABERT[33] (6-mer model), using the suggested hyperparameter settings given for fine-tuning on the method's GitHub page. We added it as an additional baseline for this task to make our benchmark more comprehensive. GenomeNet-Architect's optimized models outperform all baseline models, showing substantial improvement in pathogenicity detection (up to 11% improvement, see Fig. 4).

Additionally, we adapted the models originally optimized for the viral classification (initially optimized for sequences of 150 nt, CNN-RNN-6h, and CNN-GAP-6h) by adjusting the input size to 250 nt to evaluate how well performance of an architecture optimized for one task transfers to a different task and conditions. These architectures are renamed to have "VC" (short for viral classification) as a suffix. The comparable performance of "VC" models to models detected by GenomeNet-Architect on this dataset "GAP-CNN" and "GAP-RNN" shows good transfer between related tasks on genome data (Fig. 4).

To enhance the predictive accuracy and robustness, we explored the efficacy of ensemble approaches, a technique that combines multiple model predictions, akin to the approach presented in Bartoszewicz et al.[13], merging RC-LSTM and RC-CNN models into an RC-CNN + LSTM ensemble. Our experiments with ensemble models, including GAP + RNN-VC, GAP + RNN, and the 4-model ensemble combining both CNN-RNN and CNN-GAP variants, demonstrate a notable decrease in misclassification rates. Specifically, the 4-model ensemble reduced misclassification rates by 11% compared to the RC-CNN + LSTM baseline, with a single model improvement of 8% for CNN-GAP-VC versus RC-CNN.

## Discussion

GenomeNet-Architect defines an HP configuration search space for neural architectures that extends and generalizes successful genome data architectures from the past. This adaptable search space is coupled with an efficient black-box optimization method that can generate more optimal network architectures for genome-related tasks compared to expert-

**Fig. 4 | Comparative analysis of misclassification rates in the pathogenicity detection task.** The baseline models are shown in gray, while the red bars indicate the models developed by GenomeNet-Architect. The data for the dataset itself and the baseline results, with the exception of DNABERT[33], were derived from the DeePaC study[13]. In addition, the pre-trained DNABERT[33] model is fine-tuned on this task and added as a baseline. The graph shows individual model performance along with the improved performance archived by the ensemble approaches and highlights the superior performance of the GenomeNet-Architect models over various baselines.

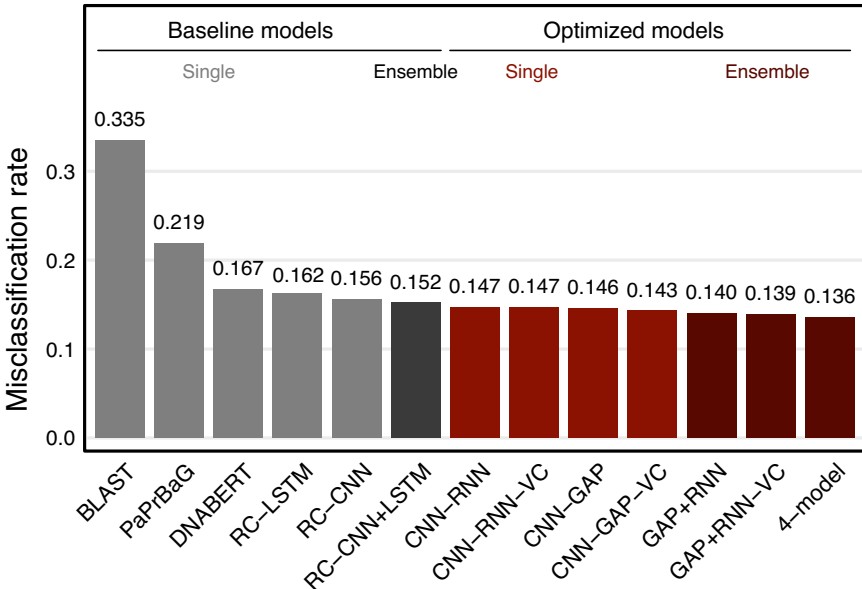

designed architectures. With GenomeNet-Architect, researchers can identify better models when applying DL to genomic datasets.

The search space used by GenomeNet-Architect leads to resulting architectures that are similar to other models used in the literature, and the individual components of the resulting architecture therefore have similar interpretations. The convolutional layer can be thought of as a pattern-matching method that encodes the presence of motifs in short sub-sequences. The global pooling layer then aggregates the information about specific patterns within the entire sequence. A global average pooling (GAP) layer measures the relative frequency of these patterns, as opposed to the encoding of the presence of individual patterns that are recorded in some models that use global max-pooling (GMP). Models using RNNs, on the other hand, are able to predict outcomes based on the spatial relationships between different patterns and can learn long-term dependencies[8,34]. The following fully connected layers are used to learn complex relationships between detected patterns and can be used in both GAP and RNN-based models[20].

The *DeepVirFinder*[6] model is an architecture that uses pooling, with a convolutional layer, followed by a global max-pooling layer and a fully connected layer. The *ViraMiner*[7] model builds on the DeepVirFinder model and proposes two branches called frequency and pattern branches, using either GAP (frequency branch) or GMP (pattern branch) after separate convolutional layers. In both branches, a fully connected layer follows, after which their output vectors are concatenated. Finally, another fully connected layer classifies whether a 300 nt sequence is human or viral DNA. Since Tampuu et al.[7] showed that the GAP alone achieves higher performance than the GMP alone, we did not include GMP in our search space. Another similarly structured architecture for viral classification is the *CHEER* model[9]. In this model, reads from 250 nt sequences are fed into four different convolutional layers with different kernel sizes: 3, 7, 11, and 15. Global max-pooling layers follow each convolutional layer, after which the paths are concatenated. Similar to other architectures, multiple fully connected layers follow the concatenation.

DeepMicrobes is another RNN-based DL model developed for viral identification. The model consists of a 12-mer embedding layer, a bidirectional LSTM layer, a self-attention layer, and several fully connected layers. The convolutional layer in this model learns local representations, and the recurrent layer can learn long-term dependencies within these local representations in a sequential manner.

Although DL models based on raw nucleotide sequences are common, there are also alternatives. One example is *Fiannaca-CNN*[10], which is a model for bacterial classification. The model uses the number of k-mer occurrences as input, which is fed into convolutional layers followed by max-pooling layers and fully connected layers. We used $k = 7$ (7-mers) in our experiments because they show that the highest accuracy is achieved using this HP. Another example is *PPR-Meta*[11], which is used to classify if the sequence is a plasmid, chromosome, or phage. The input to the model is both one-hot encoded nucleotides and 3-mers. In addition, the reverse-complements of the original inputs are concatenated to the original sequence for both inputs. The model consists of two different three convolutional layers, the outputs of which are global average pooled, concatenated, and fed into a fully connected layer. There are also max-pooling and batch-normalization layers after the first two convolutional layers.

Our HPO results provide valuable insight into the design and training of architectures for specific tasks and datasets. For example, increasing the kernel size, number of filters, and layers in convolutional networks can substantially increase both the number of trainable parameters and memory requirements, resulting in a trade-off. Many existing models, such as DeepVirFinder[6], ViraMiner[7] or CHEER[9], are limited to a single (or multiple but parallelized, not sequential) convolutional layer with a large kernel size (up to ~15) and a large number of filters ( ~ 1000). Although PPR-Meta[11] proposes a deeper model (3 sequential convolutional layers), it compensates by reducing the kernel size (down to 3). Our HPO framework has discovered an architecture that performs better on viral classification in terms of accuracy: deeper (7 convolutional layers) with a smaller number of filters in the final convolutional layers (as low as 30).

In examining model architectures that perform well across two datasets and three different sequence lengths, we sought to identify common trends and patterns in their architecture designs. Our analysis reveals that architectures with GAP layers typically incorporate more convolutional layers (5 to 7, as opposed to 1 to 5 in RNN models) and more fully connected layers (1 or 2, vs. 0 or 1) (Supplementary Table 1). The preference for GAP layers is likely due to their function in aggregating information learned by convolutional layers through averaging over the sequence, instead of learning representations by optimizing its own weights. Compared to CNN-RNN models, CNN-GAP models mainly use fully connected layers to integrate long-range information. Furthermore, LSTM layers are consistently preferred to GRU layers. Our findings also indicate a general avoidance of

multiple recurrent layers, while bidirectional RNN layers are preferred over unidirectional ones. In terms of training hyperparameters, the optimized learning rate is typically between $10^{-3}$ and $10^{-4}$ with the Adam[35] optimizer more commonly chosen over alternatives like Adagrad[36], Rmsprop, and SGD[37]. It is important to note, however, that these trends are observations and may not universally apply to every dataset or task. Therefore, we recommend running GenomeNet-Architect on the specific dataset and task in question to tailor the model architecture for optimal performance in each unique scenario.

The results of the 10,000 nt setting in contrast to the 150 nt setting are noteworthy in that there is a much smaller improvement in accuracy over the baselines in the 10,000 nt setting. This is because the viral identification task becomes "too easy" at 10,000 nucleotides, leaving little room for improvement. In such settings, where relatively simple models already perform sufficiently well, it may not be worth the considerable computational overhead of finding a specialized architecture.

## Method

### Hyperparameter search space

The hyperparameter space used for optimization is listed in Table 1 and described in more detail here.

The first part of the model constructed by GenomeNet-Architect consists of a sequence of convolutional blocks (Fig. 1), each of which consists of convolutional layers. The number of blocks ($N_{cb}$) and the number of layers in each block ($s_{cb}$) is determined by the HPs $n_{cb}$ and $n_c$ in the following way: $N_{cb}$ is directly set to $n_{cb}$ unless $n_c$ (which relates to the total number of convolutional layers) is less than that. Their relation is therefore

$$N_{cb} = \begin{cases} n_c, & \text{if } n_c \leq n_{cb} \\ n_{cb}, & \text{otherwise} \end{cases}$$

$s_{cb}$ is calculated by rounding the ratio of the $n_c$ hyperparameter to the actual number of convolutional blocks $N_{cb}$:

$$s_{cb} = round\left(\frac{n_c}{N_{cb}}\right).$$

This results in $n_c$ determining the approximate total number of convolutional layers while satisfying the constraint that each convolutional block has the same (integer) number of layers. The total number of convolutional layers is then given by

$$N_c = N_{cb} \times s_{cb}.$$

$f_0$ and $f_{end}$ determine the number of filters in the first or last convolutional layers, respectively. The number of filters in intermediate layers is interpolated exponentially. If residual blocks are used, the number of filters within each convolutional block needs to be the same, in which case the number of filters changes block-wise. Otherwise, each convolutional layer can have a different number of filters. If there is only one convolutional layer, $\lceil f_0 \rceil$ is used as the number of filters in this layer. Thus, the number of filters for the $i^{th}$ convolutional layer is:

$$f_i = \left\lceil f_0 \times \left(\frac{f_{end}}{f_0}\right)^{j(i)} \right\rceil, \quad j(i) = \begin{cases} \left\lfloor \frac{i}{s_{cb}} \right\rfloor \times \frac{1}{N_{cb}-1}, & \text{if } res\_block \\ \frac{i}{N_c-1}, & \text{otherwise} \end{cases}.$$

The kernel size of the convolutional layers is also exponentially interpolated between $k_0$ and $k_{end}$. If the model has only one convolutional layer, the kernel size is set to $\lceil k_0 \rceil$. The kernel size of the convolutional layer $i$ is:

$$k_i = \left\lceil k_0 \times \left(\frac{k_{end}}{k_0}\right)^{\frac{i}{N_c-1}} \right\rceil.$$

The convolutional layers can use dilated convolutions, where the dilation factor increases exponentially from 1 to $d_{end}$ within each convolutional block. Using "***rem***" as the remainder operation, the dilation factor for each layer is then:

$$d_i = \left\lceil d_{end}^{\left(\lfloor i\, \textbf{\textit{rem}}\, s_{cb} \rfloor\right)/(s_{cb}-1)} \right\rceil.$$

We apply max-pooling after convolutional layers, depending on the total max-pooling factor $p_{end}$. Max pooling layers of stride and a kernel size of 2 or the power of 2 are inserted between convolutional layers so that the sequence length is reduced exponentially along the model. $p_{end}$ represents the approximate value of total reduction in the sequence length before the output of the convolutional part is fed into the last GAP layer or into the RNN layers depending on the model type.

For CNN-GAP, outputs from multiple convolutional blocks can be pooled, concatenated, and fed into a fully connected network. Out of $N_{cb}$ outputs, the last $min(1, \lceil (1 - r_s) \times N_{cb} \rceil)$ of them are fed into global average pooling layers, where $r_s$ is the skip ratio hyperparameter.

### Hyperparameter optimization process

GenomeNet-Architect uses the mlrMBO software[38] with a Gaussian process model from the DiceKriging R package[39] configured with a Matérn-3/2 kernel[40] for optimization. It uses the UCB[31] infill criterion, sampling $\lambda$ from an exponential distribution as a batch proposal method[32]. In our experiment, we proposed three different configurations simultaneously in each iteration.

For both tasks, we trained the proposed model configurations for a given amount of time and then evaluated them afterwards on the validation set. For each architecture (CNN-GAP and CNN-RNN) and for each sequence length of the viral classification task (150 nt and 10,000 nt), the best-performing model configuration found within the optimization setting (2 h, 6 h) was saved and considered for further evaluation. For the pathogenicity detection task, we only evaluated the 2 h optimization. For each task and sequence length value, the first $t = t_1$ (2 h) optimization evaluated a total of 788 configurations, parallelized on 24 GPUs, and ran for 2.8 days (wall time). For the viral classification task, the warm-started $t = t_2$ (6 h) optimization evaluated 408 more configurations and ran for 7.0 days for each sequence length value.

During HPO, the number of samples between model validation evaluations was set dynamically, depending on the time taken for a single model training step. It was chosen so that approximately 20 validation evaluations were performed for each model in the first phase ($t = 2$ h), and approximately 100 validation evaluations were performed in the second phase ($t = 6$ hours). In the first phase, the highest validation accuracy found during model training was used as the objective value to be optimized. In the second phase, the second-highest validation accuracy found in the last 20 validation evaluations was used as the objective value. This was done to avoid rewarding models with a very noisy training process with performance outliers.

The batch size of each model architecture is chosen to be as large as possible while still fitting into GPU memory. To do this, GenomeNet-Architect performs a binary search to find the largest model that still fits in the GPU and subtracts a 10% safety margin to avoid potential training failures.

### Architecture evaluation and benchmarks

For the viral classification task, the training and validation samples are generated by randomly sampling FASTA genome files and splitting them into disjoint consecutive subsequences from a random starting point. A batch size that is a multiple of 3 (the number of target classes) is used, and each batch contains the same number of samples from each class. Since we work with datasets that have different quantities of data for each class, this effectively oversamples the minor classes compared to the largest class. The validation set performance was evaluated at regular intervals after training

**Table 2 | Description of the datasets used in our experiments**

| Class | Number of FASTA Files | | | Number of Sequences (L = 150) | | | Number of Sequences (L = 10k) | | |
|---|---|---|---|---|---|---|---|---|---|
| | Training | Validation | Test | Training | Validation | Test | Training | Validation | Test |
| Bacteria | 15,826 | 4523 | 2263 | 373,404,076 | 118,398,785 | 59,111,408 | 5,579,451 | 1,772,121 | 881,031 |
| Virus (non-Phage) | 18,093 | 5171 | 2588 | 2,262,526 | 568,717 | 458,526 | 20,075 | 5552 | 5350 |
| Bacteriophage | 9987 | 2855 | 1428 | 4,702,821 | 1,301,375 | 609,088 | 64,937 | 18,031 | 8400 |

on a predetermined number of samples, set to 6,000,000 for the 150 nt models and 600,000 for the 10,000 nt models. The evaluation used a sub-sample of the validation set equal to 50% of the training samples seen between each validation. During the model training, the typical batch size was 1200 for the 150 nt models, and either 120, 60, or 30 for the 10,000 nt models. Unlike during training and validation, the test set samples were not randomly generated by selecting random FASTA files. Instead, test samples were generated by iterating through all individual files, and using consecutive subsequences starting from the first position. For the pathogenicity detection task, the validation performance was evaluated at regular intervals on the complete set, specifically once after training on 5,000,000 samples. The batch size of 1000 was used for all models, except for GAP-RNN, as it was not possible with the memory of our GPU. For this model, a batch size of 500 was used.

For both tasks, we chose a learning rate schedule that automatically reduced the learning rate by half if the balanced accuracy did not increase for 3 consecutive evaluations on the validation set. We stopped the training when the balanced accuracy did not increase for 10 consecutive evaluations. This typically corresponds to stopping the training after 40/50 evaluations for the 150 nt models, 25/35 evaluations for the 10,000 nt models for the viral classification tasks, and 5/15 evaluations for the pathogenicity detection task.

To evaluate the performance of the architectures and HP configurations, the models proposed by GenomeNet-Architect were trained until convergence on the training set; convergence was checked on the validation set. The resulting models were then evaluated on a test set that was not seen during optimization.

### Datasets

For the viral classification task, we downloaded all complete bacterial and viral genomes from GeneBank and RefSeq using the genome updater script (https://github.com/pirovc/genome_updater) on 04-11-2020 with the arguments -d "genbank,refseq" -g "bacteria"/"viral" -c "all" and -l "Complete Genome". To filter out possible contamination consisting of plasmids and bacteriophages, we removed all genomes from the bacteria set with more than one chromosome. Filtering out plasmids due to their inconsistent and poor annotations in databases avoids introducing substantial noise in sequence and annotation since they can be incorrectly included or excluded in genomes. We used the taxonomic metadata to split the viral set into eukaryotic or prokaryotic viruses. Overall this resulted in three subgroups: bacteria, prokaryotic bacteriophages, and eukaryotic viruses (referred to as non-phage viruses, Table 2). To assess the model's generalization performance, we subset the genomes into training, validation, and test subsets. We used the "date of publishing" metadata to split the data by publication time, with the training data consisting mostly of genomes published before 2020, and the validation and test data consisting of more recently published genomes. Thus, when applied to newly sequenced DNA, the classification performance of the models on yet unknown data is estimated. For smaller datasets, using average nucleotide identity information (ANI) generated with tools such as Mashtree[41] to perform the splits can alternatively be used to avoid overlap between training and test data.

The training data was used for model fitting, the validation data was used to estimate generalization performance during HPO and to check for convergence during final model training, and the test data was used to compare final model performance and draw conclusions. The test data was

not seen by the optimization process. The training, validation and test sets represent approximately 70%, 20%, and 10% of the total data, respectively.

The number of FASTA files in the sets and the number of non-overlapping samples in sets of the viral classification task are listed in Table 2. Listed is the number of different non-overlapping sequences that could theoretically be extracted from the datasets, were they split into consecutive subsequences. However, whenever the training process reads a file again, e.g. in a different epoch, the starting point of the sequence to be sampled is randomized, resulting in a much larger number of possible distinct (though overlapping) samples. Because the size of the test set is imbalanced, we report class-balanced measures, i.e. measures calculated for each class individually and then averaged over all classes.

For the pathogenicity classification task, we downloaded the dataset from https://zenodo.org/records/367856313. Specifically, the used training files are nonpathogenic_train.fasta.gz, pathogenic_train.fasta.gz, the used validation files are pathogenic_val.fasta.gz, nonpathogenic_val.fasta.gz, and the used test files are nonpathogenic_test_1.fasta.gz, nonpathogenic_test_2.fasta.gz, pathogenic_test_1.fasta.gz, pathogenic_test_2.fasta.gz.

### Reporting summary

Further information on research design is available in the Nature Portfolio Reporting Summary linked to this article.

### Data availability

The data for the virus identification task is available under https://research.bifo.helmholtz-hzi.de/downloads/deepg_refpacks/architect_training_data.tar.gz. Data for the pathogen detection task is taken from a study of Bartoszewicz et al.[13] https://zenodo.org/records/3678563 as mentioned in the Datasets subsection. Source data for figures can be found in Supplementary Data 1 as well as on https://github.com/GenomeNet/Architect (https://doi.org/10.5281/zenodo.10889923).

### Code availability

The code, available at https://github.com/GenomeNet/Architect, enables users to apply the optimization process across various datasets and tasks. It is based on our R library deepG (deepg.de) and can therefore be adapted to a variety of genomics tasks that are supported by it. It uses the TensorFlow backend and can be made to run in parallel on multi-GPU-machines and compute clusters through the batchtools[42] R package.

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

## Acknowledgements

This work was funded in part by the German Federal Ministry of Education and Research (BMBF) under Grant No. 01IS18036A and under GenomeNet Grant No. 031L0199A/031L0199B and by the Deutsche Forschungsgemeinschaft DFG EXC 2155. P.C.M. received funding from the German Research Foundation (Grant number 405892038). X.-Y.T. received funding from the German Center for Infection Research (DZIF) TI BBD. C.H. received funding from NIH U19AI110820.

## Author contributions

H.A.G., M.B., and P.C.M. designed the study and experiments. H.A.G. was responsible for coding the baselines and optimized architectures, and for training, evaluating, and analyzing them. H.A.G. drafted the initial manuscript. R.M. and J.M. contributed to code development. M.B. was the main developer of the hyperparameter optimization code and the multi-fidelity approach used. M.B. also supervised and contributed to the model code, and ran the model-based optimization experiments to find optimal architectures. G.R. provided computational resources and support. E.A.F. and C.H. contributed to virus identification and benchmarking. M.R., A.C.M., B.B., and X.-Y.T. provided valuable feedback and input throughout the

project. The project was supervised by M.B. and P.C.M. All authors contributed to the writing of the manuscript.

## Funding

## Competing interests
The authors declare no competing interests.
