## [Peer Review File · Communications Biology]

Reviewers' comments:

Reviewer #1 (Remarks to the Author):

Gündüz et al. present an automated toolkit for neural architecture search and hyperparameter tuning focused on CNN-based approaches and raw DNA sequence inputs. This could be a potentially useful framework for automated training of highly performing architectures for a variety of tasks in genomics. The authors benchmarked their method against a wide variety of expert-designed architectures. The paper is very well written, with a clear motivation, meticulous presentation of the methods, and in-depth analysis of the results. However, there are some issues; addressing them would strengthen the manuscript and highlight the authors' contributions. This would include mainly clarification of some technical details, but also more explicitly describing the novelty of the proposed solution (which may not be immediately apparent), discussing some important aspects of the dataset, and, ideally, some additional benchmarking against alternative methods not considered by the authors (or, if not feasible, discussing the reasons). Overall, it is likely that the presented framework could find a community of active users in the field of microbial & viral genomics.

Major:

1. The authors write that no general-purpose MBO-based NAS frameworks focus on DNA data (lines 71-73) – could you explain why it is not possible to use the existing frameworks for genome sequence data? Also, the brief paragraph in Methods (lines 360-363) refers to other software packages and methods used for optimization, without highlighting the differences between GenomeNet-Architect and other tools. Overall, it is not fully clear if the main contribution of the manuscript is the NAS/hyperparameter optimization method, the design of the search space, the software package, or a combination of all those in a specific application context. More explicitly contrasting the previously available tools with the authors' contributions would help understand and appreciate where most novelty of the proposed approach lies.
2. Could the authors please share the list of accession IDs used in the dataset, including organism names, their ground-truth label (bacteria/phage/eukaryotic virus) and assignment to training/validation/test sets? Without this information (including the train/val/test split) the results cannot be fully reproduced.
3. In Figure 3, adding some non-ML benchmarks would help see the relative performance of the ML approaches compared to more standard alternatives. Good choices for 150nt data would be e.g. read mapping of test reads to training genomes (e.g. bwa/bowtie2) and k-mer based taxonomic classification (e.g. Kraken2). If the authors do not find these additional benchmarks to be potentially informative, a brief comment/explanation on the reasons would be very useful for the reader.
4. I fully appreciate that the authors have already benchmarked multiple expert-designed architectures. However, in Figure 3, and benchmarking in general, the authors could consider another one: in <https://doi.org/10.1093/bioinformatics/btac495>, the authors present models for read-based classification in a related task: pathogenic bacteria vs human viruses vs a joint class of non-pathogenic bacteria and non-human viruses (incl. phages); optionally also fungi. A direct comparison would be interesting as the authors of that paper report high performance (above 85%) from raw Illumina reads (albeit longer than 150nt), especially since the other baselines in Fig. 3 seem to underperform. They also have a way of scaling this up to longer sequences, up to genomes (although as noted by Gündüz et al., additional performance gains for 10kb sequences are unlikely to be significant). If the authors do not find

this additional benchmark to be potentially informative, a brief comment/explanation on the reasons would be very useful for the reader.

5. The authors use data published before 2020 for training, and later data for validation and test. However, this split does not account for sequence similarity. 'Unknown' (in the temporal sense) test data is not necessarily fully unknown, as it may include bacterial strains very similar to those included in training, phages that are very similar to those in training etc. Therefore, the 'novelty' of that data is not guaranteed. This may be ok depending on the application, but one cannot assume the models generalize to previously unseen taxa; the models can be likely reliably used only on the species/strains/viruses already used in training. For “truly novel” (e.g. highly divergent) viruses/bacteria, the performance may deteriorate drastically. This limitation should be discussed. Further, in general relying only on a temporal benchmark (which is admittedly very useful, even if not solving all problems!) risks adding additional undesirable biases (e.g. there could be some systematic differences in older vs newer data)

6. Filtering out all genomes with more than one “chromosome” sounds like a relatively drastic step – in many cases the plasmids may be an inherent part of the biology of a given species, rather than just “contamination”. E.g. especially for the 150nt models, if applied to real sequencing data, one could expect plasmid reads to be present in the dataset – they would be effectively out-of-distribution and the classification outcomes could be unreliable. Further, the classifier’s performance could be unpredictable on the removed species, which could include important pathogens. An additional comment on why this step is necessary and justified would be very helpful. Further, it would be interesting to see how many genomes like that have been removed.

7. The authors simulated 150nt reads by simply splitting the FASTA genome files into disjoint sequences, but it does not realistically model the noise caused by sequencing errors. Using a dedicated short read simulator (e.g. one of the tools benchmarked here: <https://www.nature.com/articles/s41437-022-00577-3>) would be highly recommended. A brief proof-of-concept on real sequencing data to prove that the models work on real data would be necessary to evaluate performance on real raw reads. Otherwise it may happen that real reads are OOD and the performance suffers. This is especially the case when (e.g. after trimming) not all reads are of the same length. While this may not be crucial for benchmarking of the NAS method, the level of noise could influence the results of the architecture search and hyperparameter tuning – so this should be either discussed or taken into account.

Minor:

8. The toolkit seems to be dependent strictly on tensorflow, the authors’ own library deepG and uses R – a less common choice. It would be great to explicitly mention all those dependencies and design choices, as they may directly influence whether the potential users are willing to adopt the tool.

9. The authors write that there’s no consensus on optimal architecture design in computational biology. This is absolutely true – but would one expect such a consensus in a field as broad as computational biology, with many different tasks and subfields? Considering how different the datasets are, e.g. for regulatory genomics, microbial genomics, protein design, protein structure or function predictions, MS-based proteomics/metabolomics, network-based systems biology, single-cell omics, SNPs, raw ONT data, etc., one could reasonably expect that no “consensus” will (or should) be found.

10. The authors present their approach as a generic toolkit for genomic data, but show results only for a single case study of a particular task (differentiating between bacterial, phage, and non-phage viral DNA). This is an interesting proof-of-concept and it is reasonable that the results could carry over to other applications, but since the design search space was already limited by specific design choices, this

is not guaranteed. Adding results on another task (ideally very different in terms of underlying biology) would be most convincing; alternatively, this limitation should be explicitly discussed.

11. The only task evaluated here is a classification task – since one of the goals of NAS methods is to be flexible enough to be useful for previously unexplored tasks, does the framework support also, e.g. regression or more complex tasks beyond microbial/viral genomics like predicting TF binding profiles?

12. In lines 37-41, the authors introduce two example architectures involving RNNs for genomic data, but it is not immediately clear that they are just two of multiple works using RNNs in this context, and why those two were selected. More explicitly stating that those are just examples will make it easier for a reader to get a good overview of the field.

13. In lines 125-127, the authors comment on the fact that optimizing the hyperparameter values for each layer individually would make the optimization problem more difficult. This is true; but at the same time this would allow a more comprehensive search space. Some expert-designed architectures in the field would not have been discovered by GenomeNet-Architect, including e.g. simple analogs of classic ResNet architectures for 1D sequence inputs (used e.g. by a related approach mentioned in a comment above).

14. The design of the CNN-GAP model is not trivial – the skip ratio parameter and the concatenation of outputs of different blocks are interesting design choices. At the same time, they limit the design space a bit. It would be great if the authors could explain their rationale for such a design in a bit more detail (including maybe a comment on why only the output of the last couple of layers are concatenated when skip ratio are used, rather than e.g. some other mix of early and later layers)

15. In Table 1, the authors define hyperparameters “kernel size” and “filter size”. “Convolutional filter” and “convolutional kernel” are often used as synonyms – it would help if the authors could explain what the difference is here. Also, “size” sounds like kernel/filter width – should one of those be actually the number of filters/kernels/feature detectors in a layer?

16. It is not immediately obvious why the the max pooling factor p_{end} is called “total” max pooling (this could suggest also other max pooling factors or some kind of progression)

17. In Table 1, it is not fully clear if the presented parameter values are the only ones supported by GenomeNet-Architect, or just the ones tested by the authors in their experiments. Can the user extend (or limit) the search space, e.g. allowing more or less layers/units than the authors etc.?

18. The authors write the using the UCB infill criterion and sampling its lambda-parameter from an exponential distribution is a “particularly straightforward method”. This will most likely not be obvious (or simple) for most readers, even with a computational background, and many readers of Comms Biology can be expected to have a mainly biology background instead. At the same time, it is a very important technical detail of the proposed approach. Please briefly explain what UCB stands for, how does using the UCB infill criterion work, and what the effect of the lambda parameter is; in general how this method works. Please make it understandable for readers of different backgrounds.

19. While differentiating between bacteria, phage and non-phage viral DNA can be argued to indeed be a taxonomic classification task, it works only on an extremely high level, while the reader could expect “taxonomic classification” to mean handling multiple hierarchical taxonomic units and multiple levels (down to individual species or even strains). It would be good to make this distinction more explicit.

20. The authors write that GMP (global max pooling) alone was shown to be worse GAP alone, but this usually depends on the task at hand. This may be the case for viral & taxonomic classification tasks, but could conceivably not hold for e.g. certain problems in regulatory genomics. Would it be possible to have GMP in GenomeNet-Architect as well and simply disable it when the user suspects it's not a good idea

(as e.g. in the task selected by the authors)?

21. It is a bit surprising that Fiannaca-CNN performs so well, as one of the main strength of CNNs is translation invariance, while the k-mer occurrences in Fiannaca seem to be basically tabular data (the order of k-mer occurrence columns can usually be rearranged freely without loss of information). A filter (interpreted as a sliding window), operating on tabular data without the spatial structure that images or raw sequences have effectively has to deal with input having completely different 'meaning' depending on the filter's position. A comment on why this works nevertheless and what are the consequences of this working would be nice.

22. I really appreciate how the authors aimed to analyze some trends in the optimized design choices (lines 301-308). It would be even more convincing if that analysis included some kind of a quantitative component, e.g. some measures of statistical significance supporting the claims. Otherwise, a comment explaining that the observed trends are not necessarily significant could help the reader critically assess the proposed interpretation.

23. In Table 3, it is not fully clear if the numbers present the numbers of non-overlapping reads/fragments that can be theoretically generated (this seems to be the case), or the number or reads/fragments actually used to train the models (which could be higher, as mentioned, but potentially also lower). Making the distinction even more explicit would help.

Typos

1. In lines 191-192 “eukaryotic viruses (referred to as (bacterio-)phage) and prokaryotic viruses (referred to as viral non-phage DNA)” – should be the other way around

Reviewer #2 (Remarks to the Author):

In this manuscript, the authors present a novel framework designed to optimize neural network architectures for genomic tasks, including hyperparameter optimization. They demonstrate the efficacy of this approach using a specific viral classification task, resulting in improved prediction accuracy and reduced model size, which suggests potential downstream benefits in computational load. The development of such a method is undoubtedly innovative and holds immense potential for the field. To further enhance its practical utility, it is important to consider factors such as ease-of-use, which encompasses deployment, reproducibility, and documentation. Another critical aspect to examine is the computational burden this method imposes during the initial training phase. With these considerations in mind, I will now proceed with the evaluation of the manuscript.

L29-41 (minor): To efficiently perform NAS ... two fully connected layers.

I feel that this paragraph falls a little short from being a full review of methods that use DL in genomics, while also being very specific on a couple of niche subfields. I would suggest to add reference to a few general reviews of DL in genomics or specific subfields instead of talking about specific methods. Your method has the potential to be applied on many fields of genomics (and transcriptomics) so you are doing your manuscript a disservice with focusing too closely on one specific application.

L71-73 (minor): No general purpose MBO-based NAS framework ... 2D image data instead

I understand that this is the crux of the importance of this paper. I spent most of the introduction wondering “why not just use a generic NAS”. I would suggest to either put a mention of this in the abstract, and/or when you talk about different types of NNs and modalities of data (first paragraph) to drive home the point that genomic data is different from image or language data (and how). It will give more depth to the exploration of the question at hand.

L96-98 (major): However, the resulting architecture can also be used for other tasks that are similar to the task for which it was optimized. It is therefore possible to perform a single optimization run to solve multiple genome sequence DL tasks.

I find this counter-intuitive in my experience with training neural networks for genomics tasks. It is uncommon that hyperparameters and architecture will be similar for conceptually similar tasks when training on different datasets. I would like to see at least some proof / demonstration of this sentence esp. since it is in the Results section.

L128 (minor):

You use interpolation for the hyperparameters between first and last layer. Even though I believe you based on practical experience that this works fine, it could be worthwhile to actually show that it does. You could possibly do one run on a dataset with interpolation and one with finding the HP for intermediate layers and report the differences in training time and HP prediction.

L140 (minor):

I am looking at the architecture HP search space table, and I fail to see Dropout for the CNN layers. Isn't that a commonly used step in CNNs? Would it make sense in your proposed architectural search space?

L189 (no comment):

I think that the showcase on taxonomic classification is nice, and it really drives home the message that this type of approach can actually improve predictive values.

Discussion (major):

The whole discussion section is hyperfocused on the specific task that was used to showcase the method. I think that this does the manuscript a disservice. I would rather focus on many different potential uses of the method in various genomic and transcriptomic tasks. Since the method is generic, I believe that it has potential for many more applications.

I would also like to see some discussion regarding how easy it is to expand the method to different types of neurons (e.g. GRU) and more complicated architectures (e.g. multiple input branches such as sequence, conservation) How easy is it to deploy and use. What type of equipment is needed to run the method at an acceptable timeframe.

Overall, I believe that the method is useful and should be well used, but I think that the authors should expand their scope a little bit outside their specific expertise in phylogenetic classification.

Reviewer #3 (Remarks to the Author):

The authors proposed a neural architecture design framework (GenomeNet-Architect) for automatically

optimizing deeplearning models for genome sequence data. Given a fast accumulation of sequencing data, having an automatic architecture optimization tool is helpful. The method is clearly described. However, its utility needs to be evaluated more rigorously to support the authors' conclusions. Below please find my detailed comments.

1. The author claimed that GenomeNet-Architect is a neural architecture design framework that researchers can use to automatically optimize learning models for genome sequence data. However, the framework is only applied to one task and thus this conclusion is not well-supported. Even for virus analysis, there are several different tasks, such as taxonomic classification and protein function annotation. Different tasks have different challenges, and the distributions of the data can vary a lot. If GenomeNet-Architect is a general framework, the author should show some benchmarks on at least two tasks. Considering that this can automatically optimize the architecture, this should be feasible.

2. The author should also compare it to other frameworks, such as DNABERT [1].

[1] DNABERT: pre-trained Bidirectional Encoder Representations from Transformers model for DNA-language in genome

3. It is not clear how the authors generate the short/contig sequences. It is better to show contigs of different lengths (e.g. 2000, 5000 etc.).

4. More detailed description of how the authors used the baseline methods should be provided. In the manuscript, the authors mentioned that they collected the data by themselves and split the data by time (before or after 2020). However, the mentioned baseline methods may be trained on different datasets based on different data collection and partition methods. The authors should retrain these baseline models on the same datasets for a fair comparison.

5. As described in Section Discussion, DeepMicrobes and CHEER are taxonomic classification tools. PPR-Meta is used to classify phages, chromosomes, and plasmids. Since their original design is not optimized for the virus identification task, the comparison may not be fair. At least re-training and adaption of these baseline models should be conducted. In addition, the author should include more virus identification tools for comparison if they focus on virus identification task.

6. According to Table. 3. There are many more bacterial short/contig sequences in the datasets. How do the authors handle the data imbalance problem? The performance seems only for "balanced data". Is the test data or the training data balanced? In real applications (such as metagenomic data), the label will not be balanced at all. How does this "balanced" performance translate to the real applications?

7. More experiments should be established to demonstrate the robustness of the model. For example, how sequence similarity will affect the performance of the model. What is the performance of the real sequencing data?

Minor:

Line 293: “performs better on taxonomic classification”. There is no taxonomic classification result (e.g. family, genus, etc.) found in the manuscript. The task is more similar to virus identification.

Responses to Reviewer Feedback

Reviewer #1

We thank Reviewer #1 for the valuable comments and interest in our manuscript. The additional feedback was very helpful in improving the work, and as outlined below, we hope to have addressed your comments and edited the manuscript according to your suggestions.

Q1: The authors write that no general-purpose MBO-based NAS frameworks focus on DNA data (lines 71-73) – could you explain why it is not possible to use the existing frameworks for genome sequence data? Also, the brief paragraph in Methods (lines 360-363) refers to other software packages and methods used for optimization, without highlighting the differences between GenomeNet-Architect and other tools. Overall, it is not fully clear if the main contribution of the manuscript is the NAS/hyperparameter optimization method, the design of the search space, the software package, or a combination of all those in a specific application context. More explicitly contrasting the previously available tools with the authors' contributions would help understand and appreciate where most novelty of the proposed approach lies.

The manuscript's novelty primarily lies in its *search space* (the space of configurations that it tries out) and *software package*, GenomeNet-Architect. We address the distinct requirements of genome sequence data, which are not adequately catered to by existing neural architecture search (NAS) methods. Other methods are predominantly developed for more popular data types such as image data (as can be seen from the fact that most NAS benchmarks, listed in <https://www.automl.org/nas-overview/nasbench/>, evaluate network performance on image data), and often only focus on architecture alone, necessitating separate hyperparameter optimization. In contrast, our method simultaneously addresses hyperparameter optimization and architecture search, optimizing elements such as learning rate, dropout rate, optimizer, and filter size.

A key contribution is the specialized search space for hyperparameters, tailored to DNA sequence data. This space includes architectures commonly found in the relevant literature, like the integration of recurrent layers on top of convolutional ones, a design not typical for image data but effective for our datasets. This approach, combined with a multi-fidelity evaluation method that prioritizes shorter runs, distinguishes our method from general black-box optimization frameworks. While other frameworks can be adapted with a custom search space to DNA data as well, this would essentially be duplicating our work -- GenomeNet-Architect specifically provides this tailored space and integrates it with our MBO+NAS method.

We have revised the last paragraph of the introduction to make the difference from other NAS frameworks more clear:

"Unlike other general-purpose NAS frameworks, GenomeNet-Architect uses a search space specifically for genome data. It is made up of neural architectures and HP setups that build on

top of and generalize various architectures for the genome data that have been successfully applied in the past. This approach allows us to efficiently explore a large space of possible network architectures and identify those that perform well for genome-related tasks, creating architectures that outperform expert-guided architectures. Our method can be used for a variety of DL tasks on genome sequence data, such as genome-level, loci-level, or nucleotide-level classification and regression."

Q2: *Could the authors please share the list of accession IDs used in the dataset, including organism names, their ground-truth label (bacteria/phage/eukaryotic virus) and assignment to training/validation/test sets? Without this information (including the train/val/test split) the results cannot be fully reproduced.*

Thank you for your feedback. In response to your query, we have now provided this information available for access at the following URL: https://research.bifo.helmholtz-hzi.de/downloads/deepg_refpacks/architect_training_data.tar.gz which we also included in the Manuscripts method section. This dataset includes the FASTA files of train/test/validation splits used for the bacteriophage/eukaryotic virus task. Additionally, we have incorporated this information into the manuscript to facilitate full reproducibility of our results.

The training files for the optimization process (pathogenicity classification task) we did in line to satisfy reviewer comments are public and can be accessed which is downloadable via <https://zenodo.org/records/3678563>. Specifically, the used training files are nonpathogenic_train.fasta.gz, pathogenic_train.fasta.gz, the used validation files are pathogenic_val.fasta.gz, nonpathogenic_val.fasta.gz, and the used test files are nonpathogenic_test_1.fasta.gz, nonpathogenic_test_2.fasta.gz, pathogenic_test_1.fasta.gz, pathogenic_test_2.fasta.gz, which we now also mention in the Methods section of the manuscript.

Q3: *In Figure 3, adding some non-ML benchmarks would help see the relative performance of the ML approaches compared to more standard alternatives. Good choices for 150nt data would be e.g. read mapping of test reads to training genomes (e.g. bwa/bowtie2) and k-mer based taxonomic classification (e.g. Kraken2). If the authors do not find these additional benchmarks to be potentially informative, a brief comment/explanation on the reasons would be very useful for the reader.*

In response to the suggestions for non-ML benchmarks, we have incorporated two alignment-based baselines using Bowtie2 for short read sized data (150nt) and Minimap2 for 10k length (see **Supplementary Table 2**). In the former scenario, we align all subsequences from the test set with size 150 against an indexed Bowtie2 databases which was generated from the combined training samples (all 3 groups) using the bowtie2 setting "-N 1" allowing more mismatches in the seed region, which can lead to more alignments being found, especially in cases of slight sequence variations. Our results with this benchmark indicated a balanced accuracy of 71.7% for 150 nt sequences, which is lower than our models and the Fiannaca

baseline. We also created a new baseline based on Kraken2¹. For this, we have created a custom Kraken2 database with only the 3 taxons and the training set and used this database to assign taxons to the test dataset (see **Supplementary Table 2**).

Supplementary Table 2: Balanced accuracy of baselines for the viral classification task, including non-machine learning baselines. The optimized GenomeNet-Architect model outperforms all baseline models. The bold values represent the best models.

Model/Baseline	Balanced Accuracy	Length (nt)
Bowtie2 ²	71.7%	150
Kraken2 ¹	76.0%	
Best ML baseline (Fiannaca ³)	75.0%	
Optimized model	79.8%	
Kraken2 ¹	89.4%	10k
Best ML baseline (Fiannaca ³)	98.6%	
Optimized model	98.8%	

Q4: I fully appreciate that the authors have already benchmarked multiple expert-designed architectures. However, in Figure 3, and benchmarking in general, the authors could consider another one: in <https://doi.org/10.1093/bioinformatics/btac495>, the authors present models for read-based classification in a related task: pathogenic bacteria vs human viruses vs a joint class of non-pathogenic bacteria and non-human viruses (incl. phages); optionally also fungi. A direct comparison would be interesting as the authors of that paper report high performance (above 85%) from raw Illumina reads (albeit longer than 150nt), especially since the other baselines in Fig. 3 seem to underperform. They also have a way of scaling this up to longer sequences, up to genomes (although as noted by Gündüz et al., additional performance gains for 10kb sequences are unlikely to be significant). If the authors do not find this additional benchmark to be potentially informative, a brief comment/explanation on the reasons would be very useful for the reader.

We assume this question suggests both a new expert-designed architecture that we could compare against, as well as a different dataset that we can use to evaluate model performance:

Q4a: [...] read-based classification in a related task: pathogenic bacteria vs human viruses vs a joint class of non-pathogenic bacteria and non-human viruses (incl. phages); optionally also fungi

In response to **Q4a**, we have now added a new benchmark using the dataset <https://zenodo.org/records/3678563>, which is similar to the one outlined by **Reviewer 1**. See also our detailed response to **Reviewer 2 Q7**.

Q4b: [...] the authors have already benchmarked multiple expert-designed architectures [...] could consider another one: in <https://doi.org/10.1093/bioinformatics/btac495>

In response to **Q4b**, we now included the evaluation of a model closely resembling the one described in the suggested reference Bartoszewicz et al. ⁴, specifically utilizing an RC-equivariant ResNet-18 model. We evaluated this model on the viral classification task for both 150 nt and 10,000 nt sequences. Our benchmarks indicate that the top-performing models identified by GenomeNet-Architect exhibit an 9.6 percentage point (pp) increase in balanced accuracy for 150 nt samples (79.8% for our optimized model versus 70.2% for the RC-ResNet-18 model) and a 0.8pp increase for 10,000 nt sequences (98.8% vs. 98.0%). These differences between performances are considerable despite the RC-ResNet-18 having almost twice as many parameters as the models found by our optimization framework.

The reason we chose this very similar model instead of the exact same one is the usage of several different custom layers in the original model, which makes it considerably harder to implement, while we also note a downside of the specific model used in <https://doi.org/10.1093/bioinformatics/btac495>. Specifically, we used normal layers instead of custom reverse-complement (RC) layers in the proposed architecture while other hyperparameters of the architecture and structure are kept unchanged. For RC-equivariance, we also fed the reverse-complement sequences to the model with the same weights along with the original sequences and averaged the results during training and testing. In this way, we used the same ResNet-18 architecture in this paper with the “Siamese RC-architecture” used in the “DeePaC”⁵ study of the same team, which was also the way we used in our architectures that utilize the RC sequences.

While we acknowledge that the model we evaluated differs slightly from the original recommendation, it's worth noting that current research, such as the manuscript "Towards a Better Understanding of Reverse-Complement Equivariance for Deep Learning Models in Regulatory Genomics"⁶, suggests that the performance difference between the models with custom layers (RC parameter sharing model) and model with averaged outputs for both original and reverse-complement sequences ("Conjoined-trained" or "Siamese RC-architecture" model) is usually small. Additionally, these layers do not necessarily improve the performance and sometimes they even decrease it. Therefore we think it is very unlikely that the proposed architecture in the mentioned paper would be a better architecture than the architectures our method finds, as the baseline we tested performs substantially worse than our models.

Updated Figure 3: Predictive performance and characteristics of models found by GenomeNet-Architect and various baselines on the viral classification task. This version of the figure includes the RC-ResNet-18 model. Note: We have also adapted the batch size from the default value of 32 to the largest fitting batch size value for the experiments to be more efficient which also changed the time estimates for all models. This has no influence on prediction performance results besides changed inference time.

Q5: The authors use data published before 2020 for training, and later data for validation and test. However, this split does not account for sequence similarity. 'Unknown' (in the temporal sense) test data is not necessarily fully unknown, as it may include bacterial strains very similar to those included in training, phages that are very similar to those in training etc. Therefore, the 'novelty' of that data is not guaranteed. This may be ok depending on the application, but one cannot assume the models generalize to previously unseen taxa; the models can be likely reliably used only on the species/strains/viruses already used in training. For "truly novel" (e.g. highly divergent) viruses/bacteria, the performance may deteriorate drastically. This limitation should be discussed. Further, in general relying only on a temporal benchmark (which is admittedly very useful, even if not solving all problems!) risks adding additional undesirable biases (e.g. there could be some systematic differences in older vs newer data)

The main contribution of your manuscript is *GenomeNet-Architect*, a tool for optimizing models on various datasets. The benchmarks demonstrate its efficacy by increased model performance relative to un-optimized and baseline models. The optimized classifiers serve as evidence of the successful optimization strategy rather than as practical prediction tools. Most efforts were focused on refining the optimization process.

Our approach to dataset splitting, where we use date as a proxy for genome similarity, aims to ensure more recent genomes are included in the test set rather than the training set. While not perfect, it is an improvement over random splits that are typically performed that don't consider dataset similarities at all. However, we tested more complex methods for assessing similarity, like Mashtree⁷ but they have proven to be impractical due to scalability issues, as evidenced by the failure to run it on machines even with having over 2TB of memory.

We agree that we could use plasmid sequences as OOD samples and see how the models are handling these. Since methods like Mashtree failed for this dataset, we quantified model performance as a function of nucleotide differences in the Bowtie2 alignment step. Here we processed the BAM files to count the nucleotide similarity of sequences mapping from the test dataset to the Bowtie2 database consisting of samples from the training dataset. We binned the samples based into the following groups: exact match (alignment) of a subsample from the test set to the combined training dataset (all groups); 1-5, 6-10, 11-15, and 16-20, and differences of more than 21 nucleotides (nt), respectively. The final bin denotes samples from the test set that had no alignment at all to the training database, denoted as not aligned (NA) group. We processed these sets from the test datasets using methods to the best-performing baseline Fiannaca and to the optimized method CNN-GAP-6h. While by definition, all samples in the NA group fail to be processed by the alignment-based baseline, our optimized model CNN-GAP-6h outperformed the best-performing baseline model Fiannaca³ by 8.6 percentage points. Furthermore, our optimized model showed a better performance compared to Fiannaca³ in all bins. When tested simulated reads that have an Illumina error profile, our optimized models outperform Fiannaca by 10.5 percentage points. We added this information to the manuscript:

*“We also analyzed the performance of the optimized model stratified by the degree of genomic differences to the training data. Our findings show that it consistently outperforms the Fiannaca baseline, demonstrating that GenomeNet-Architect’s performance does not come from overfitting on sequences that occur in the training dataset (**Supplementary Figure 1**).”*

We also added the following information to the "Datasets" section:

“For smaller datasets, using average nucleotide identity information (ANI) generated with tools such as Mashtree⁷ to perform the splits can alternatively be used to avoid overlap between training and test data.”

Supplementary Figure 1: Model performance stratified similarity groups based on the closest sample in the training set via Bowtie2² alignment.

Q6: *Filtering out all genomes with more than one “chromosome” sounds like a relatively drastic step – in many cases the plasmids may be an inherent part of the biology of a given species, rather than just “contamination”. E.g. especially for the 150nt models, if applied to real sequencing data, one could expect plasmid reads to be present in the dataset – they would be effectively out-of-distribution and the classification outcomes could be unreliable. Further, the classifier’s performance could be unpredictable on the removed species, which could include important pathogens. An additional comment on why this step is necessary and justified would be very helpful. Further, it would be interesting to see how many genomes like that have been removed.*

In response to the reviewer's concerns, we acknowledge the significance of plasmids in certain species' biology. However, for our study's scope, the dataset's selection is secondary to our main objective (as outlined in our answer to **Q5** above), which is comparing relative improvements across re-trained models. We chose to exclude plasmids primarily due to their inconsistent and poor annotations in databases, which can introduce substantial noise in sequence and annotation since they can be incorrectly included or excluded in genomes.

To address the reviewer's query on the impact of this exclusion, we analyzed plasmids from PLSDB with our optimized models and the Fiannaca method, using both 150nt and 10k nt sequences. Results show that our models categorize more plasmid samples as bacteria than Fiannaca, particularly in the 10k nt category (96.4% vs. 86.8%), suggesting effective processing of these out-of-distribution (OOD) samples. However, the 150 nt results are less definitive, with frequent misclassifications as viruses or phages. We included this information in the manuscript:

“Filtering out plasmids due to their inconsistent and poor annotations in databases avoids introducing substantial noise in sequence and annotation since they can be incorrectly included or excluded in genomes”

***Q7:** The authors simulated 150nt reads by simply splitting the FASTA genome files into disjoint sequences, but it does not realistically model the noise caused by sequencing errors. Using a dedicated short read simulator (e.g. one of the tools benchmarked here:<https://www.nature.com/articles/s41437-022-00577-3>) would be highly recommended. A brief proof-of-concept on real sequencing data to prove that the models work on real data would be necessary to evaluate performance on real raw reads. Otherwise it may happen that real reads are OOD and the performance suffers. This is especially the case when (e.g. after trimming) not all reads are of the same length. While this may not be crucial for benchmarking of the NAS method, the level of noise could influence the results of the architecture search and hyperparameter tuning – so this should be either discussed or taken into account.*

In response to the Reviewer's suggestion, we performed a benchmark using simulated reads using the Illumina error profile (using NGSNGS⁸) using genomes from the test dataset using the same setting as in **Q5** (binned by read similarity). Our model outperforms the best baseline in all bin intervals consistently, while the performance difference between models is higher for test samples that are less similar to the training set. Specifically, our model predicts 67.2% balanced accuracy for test samples that do not match the samples in the training set, while the best baseline performance is 58.5%.

We updated the manuscript to include this finding in the result section which reads: *“The superior performance persists even when reads are simulated with the Illumina read error profile, underscoring the tool's effectiveness across diverse genomic sequencing challenges (Supplementary Figure 2).”*

Q8: *The toolkit seems to be dependent strictly on tensorflow, the authors' own library deepG and uses R – a less common choice. It would be great to explicitly mention all those dependencies and design choices, as they may directly influence whether the potential users are willing to adopt the tool.*

We appreciate your feedback regarding the dependencies and design choices of our toolkit. It's important to note that R is a popular language in the biological research community, which we anticipate will encourage its adoption. The decision to use R was also influenced by the integration of our deep learning library, *deepG* ([deepg.de](https://github.com/GenomeNet/Architect)), which is written in R. This compatibility allows *GenomeNet-Architect* to leverage *deepG*'s capabilities⁹. Furthermore, our ongoing projects like *Self-GenomeNet*¹⁰ are also based on the same R codebase, ensuring consistency across our tools¹⁰.

We acknowledge that the current version of *GenomeNet-Architect* is tailored specifically for TensorFlow models. This could at first seem to be a limitation for users preferring other frameworks. However, TensorFlow's significant market share and widespread use in the scientific community justify its choice. It is furthermore entirely possible to have both TensorFlow and (Py)Torch installed at the same time, so *GenomeNet-Architect* can be used by users who evaluate other experiments with Torch.

We included a mention of the dependencies in the **Code Availability** statement which reads:

“The code available at <https://github.com/GenomeNet/Architect>, which is based on our R library deepG (deepg.de) and the TensorFlow backend, enables users to apply the optimization process across various datasets and tasks.”

Q9: *The authors write that there's no consensus on optimal architecture design in computational biology. This is absolutely true – but would one expect such a consensus in a field as broad as computational biology, with many different tasks and subfields? Considering how different the datasets are, e.g. for regulatory genomics, microbial genomics, protein design, protein structure or function predictions, MS-based proteomics/metabolomics, network-based systems biology, single-cell omics, SNPs, raw ONT data, etc., one could reasonably expect that no “consensus” will (or should) be found.*

We agree that the diversity of datasets in computational biology, encompassing regulatory and microbial genomics, protein design, structure, and function predictions, MS-based proteomics/metabolomics, network-based systems biology, single-cell omics, SNPs, and raw ONT data, inherently negates the feasibility of a universal, one-size-fits-all architectural solution. This is why we think *GenomeNet-Architect* is important. Our manuscript does not claim to have discovered a singular architecture that can be universally applied across all tasks. Instead, we emphasize the necessity of tailored optimization for specific datasets or modeling tasks, underlining the diverse nature of the field. Our work aims to highlight the importance of customization and optimization in computational models to address the unique challenges

presented by each dataset, reinforcing the concept that successful application in computational biology requires a nuanced, data-specific approach.

However, we argue that, despite this diversity, certain domains within computational biology, such as genomics, have a degree of similarity that might not be as pronounced in the comparison between vastly different fields like image and speech recognition. This relative homogeneity within nucleotide-level tasks suggest that while a universal architecture might be elusive, a more target approach to optimization could yield effective solutions within a narrower domain. Also please see our related answer to **Reviewer 2 Q3**.

***Q10:** The authors present their approach as a generic toolkit for genomic data, but show results only for a single case study of a particular task (differentiating between bacterial, phage, and non-phage viral DNA). This is an interesting proof-of-concept and it is reasonable that the results could carry over to other applications, but since the design search space was already limited by specific design choices, this is not guaranteed. Adding results on another task (ideally very different in terms of underlying biology) would be most convincing; alternatively, this limitation should be explicitly discussed.*

In response to **Q10**, we have addressed this concern by incorporating additional datasets into our analysis, as detailed in our answer to **Reviewer 2 Q7**. This inclusion serves to broaden the applicability of our findings and reinforce the versatility of our approach.

Note that a larger search space involves computational trade-offs. A search space that enumerates all possible neural architectures without constraints could contain an architecture more fitting for a specific problem, but optimizing on this larger search space until that architecture is found will take much longer and is challenging with today's compute resources. Our current approach balances the depth of the search with these practical constraints, aiming to provide a robust yet feasible solution for architecture optimization in genomic sequence analysis. We plan to extend our search space and refine our optimization techniques to accommodate a wider variety of tasks and data complexities in follow up studies.

***Q11:** The only task evaluated here is a classification task – since one of the goals of NAS methods is to be flexible enough to be useful for previously unexplored tasks, does the framework support also, e.g. regression or more complex tasks beyond microbial/viral genomics like predicting TF binding profiles?*

Our method can work with various genome-related tasks where predictions are made for genomic data -- these can be classification or regression tasks. We have now adjusted the library to make configuring the task easier. The process for this is documented by the GenomeNet-Architect package. We have modified the **Discussion** section of the manuscript to highlight this possibility which reads: “*Our method can be used for a variety of DL tasks for genome sequence data, such as genome-level, loci-level, or nucleotide-level classification and regression.*”

Q12: *In lines 37-41, the authors introduce two example architectures involving RNNs for genomic data, but it is not immediately clear that they are just two of multiple works using RNNs in this context, and why those two were selected. More explicitly stating that those are just examples will make it easier for a reader to get a good overview of the field.*

Thanks for the suggestion, we have now modified it as follows, clearly mentioning that there are numerous works in genomics to use RNN layers and added more citations in this regard which reads:

“While numerous works in genomics use RNN layers^{11–15}, one example is *Seeker*¹⁶, an RNN-based model that employs an LSTM layer for bacteriophage detection. Furthermore, by stacking them sequentially, integrating convolutional and recurrent layers enhances model capability. For instance, the model developed by Wang et al.¹⁷ demonstrates this approach by placing an RNN on top of a convolutional layer, followed by two fully connected layers. A similar configuration is utilized in the DanQ model¹⁸, showcasing the effectiveness of combining recurrent and convolutional layers.”

Q13: *In lines 125-127, the authors comment on the fact that optimizing the hyperparameter values for each layer individually would make the optimization problem more difficult. This is true; but at the same time this would allow a more comprehensive search space. Some expert-designed architectures in the field would not have been discovered by GenomeNet-Architect, including e.g. simple analogs of classic ResNet architectures for 1D sequence inputs (used e.g. by a related approach mentioned in a comment above).*

As described in our answer to **Q10**, there is a tradeoff between the comprehensiveness of the search space and the computational resources necessary to search through it. Introducing more hyperparameters to be optimized in this problem would, in particular, negatively affect the efficacy of the optimization algorithm being used here: Bayesian optimization works best when it only handles a moderate number of dimensions¹⁹. We have therefore designed the search space to cover the aspects well known to have a large influence on network performance (e.g. hyperparameters of the optimizer, width and depth of the network).

While it is true that not all expert-designed architectures can be discovered by our method, there are close architectures that are within our search space. As an example, consider the the following hypothetical hyperparameter configuration (**Rebuttal Table 1**), which is close to the mentioned ResNet architecture for 1D input⁴ with kernel sizes of 7, 5, 5, ..., (16 times 5) and number of filters set to 64, 64, ..., 128, 128, ..., 512 (5 times 64, 4 times 128, 256, and 512) :

Rebuttal Table 1: An architecture configuration close to ResNet which is in the search space and could potentially be explored by GenomeNet-Architect.

Parameter	Value
First Kernel Size (k_0)	5
Last Kernel Size (k_{end})	5
Number of Conv. Layers (n_c)	16
First Number of Filters (f_0)	64
Last Number of Filters (f_{end})	512
Number of Conv. Blocks (n_{cb})	4
Residual Block	TRUE
Skip Ratio for GAP (r_s)	0.9
Number of Dense Layers	0
Model Type	GAP

These two architectures have the same values in terms of kernel size and number of filters for 16 layers out of 17 convolutional layers in the mentioned ResNet analog. On top of this, both architectures have residual connections, a global average pooling layer applied on top of the final convolutional layer, and a fully connected layer on top of the global average pooling layer. Therefore we believe similar architectures to the mentioned model can indeed be discovered by our architecture.

Besides being able to explore similar models, our experiments show better models are discovered in practice for a given dataset than expert-designed models. For example, we show experimentally that the mentioned ResNet analog model in one of your other questions performs considerably worse than architectures that the Architect detects, despite having a higher number of parameters. (Class balanced accuracy of 79.78% and 98.78% for 150 and 10,000-length sequences respectively for GenomeNet-Architect, while the RC-ResNet-18 model achieves 70.21% and 97.99% respectively.)

Q14: *The design of the CNN-GAP model is not trivial – the skip ratio parameter and the concatenation of outputs of different blocks are interesting design choices. At the same time, they limit the design space a bit. It would be great if the authors could explain their rationale for such a design in a bit more detail (including maybe a comment on why only the output of the last couple of layers are concatenated when skip ratio are used, rather than e.g. some other mix of early and later layers)*

In the literature, it is common to use models that work similar to the CNN-GAP model (including the mentioned ResNet model by the reviewer in **Q4** and **Q13**), but that only feed the output of the very last convolutional layer to a GAP layer (corresponding to a large value of the skip ratio parameter r_s). We also believe that it makes sense to include the output from earlier layers (small r_s values). We chose to express the number of layers whose output is included as a continuous value between 0 and 1 so that it is a value independent of the total number of convolutional blocks (n_{cb}).

By varying the number of convolutional blocks while keeping the number of CNN layers constant, it is possible, to a degree, to vary the specific indices of intermediate layers whose output is collected for the GAP layer. As an example, for a GAP model with 12 layers and skip block ration of 0.2, if the number of convolutional blocks is 3, then the output of 4th, 8th and 12th convolutional layers are global average pooled, concatenated and fed into the fully connected layers. We therefore think that having a more complicated way of specifying the positions within the network from which to collect outputs would not give the benefit that would justify this complication.

We would also note that the output of the GAP layer goes through one or more linear layers with trained weights -- should it turn out that the output of very specific intermediate layers is more "interesting" than others, this fact could still be found during neural network training by adjusting their weights.

Q15: *In Table 1, the authors define hyperparameters “kernel size” and “filter size”. “Convolutional filter” and “convolutional kernel” are often used as synonyms – it would help if the authors could explain what the difference is here. Also, “size” sounds like kernel/filter width – should one of those be actually the number of filters/kernels/feature detectors in a layer?*

The “filter size” was referring to the number of kernels. We came up with this name, because on Tensorflow, the argument for the number of kernels is called “filters”, whereas kernel_size argument refers to the kernel size and we use this name as is. Nevertheless, we understand that it might be confusing, therefore we have now changed the “filter size” to “number of filters”.

Q16: *It is not immediately obvious why the max pooling factor p_{end} is called “total” max pooling (this could suggest also other max pooling factors or some kind of progression)*

It is called total max-pooling because it determines the approximate value of the reduction in the size of the data, which is achieved by using max-pooling layers of 2 (both kernel size and stride value) or powers of 2. For example, when the input sequence length is 10,000 and the “total max-pooling” is 4, typically 2 max-pooling layers with kernel size and stride value of 2 are applied after different layers to reduce the length of the output of the last convolutional layer to 2,500.

We now added this text to the manuscript: *“Max pooling layers of stride and kernel size of 2 or the power of 2 are inserted between convolutional layers so that the sequence length is reduced exponentially along the model. p_{end} represents the approximate value of total reduction in the sequence length before the output of the convolutional part is being fed into the last GAP layer or into the RNN layers depending on the model type.”*

Q17: *In Table 1, it is not fully clear if the presented parameter values are the only ones supported by GenomeNet-Architect, or just the ones tested by the authors in their experiments. Can the user extend (or limit) the search space, e.g. allowing more or less layers/units than the authors etc.?*

The specific search space presented here is one of our main contributions: it was specifically designed to cover typical network configurations used for deep learning on genomic data, to be economical in the number of (costly) search space dimensions while allowing the discovery of novel architectures. However, with our provided software it is relatively easy to specify different parameter ranges, and even possible to use an entirely different search space. We have updated results section to highlight this possibility:

“While the provided search space is our recommendation, our method also supports defining a custom search space, e.g. allowing more layers, or including GMP instead of GAP.”

Q18: *The authors write the using the UCB infill criterion and sampling its lambda-parameter from an exponential distribution is a “particularly straightforward method”. This will most likely not be obvious (or simple) for most readers, even with a computational background, and many readers of Comms Biology can be expected to have a mainly biology background instead. At the same time, it is a very important technical detail of the proposed approach. Please briefly explain what UCB stands for, how does using the UCB infill criterion work, and what the effect of the lambda parameter is; in general how this method works. Please make it understandable for readers of different backgrounds.*

We extended the paragraph where we introduce UCB to explain the method in some more detail. We also stress that our choice of using UCB for multi-point proposal is a particularly simple method that still performs very well in benchmark comparisons. The paragraph is now:

“GenomeNet-Architect parallelizes the HP tuning process across multiple GPUs to reduce the overall optimization time. There are a variety of multi-point proposal techniques that allow MBO

to evaluate several different HP configurations simultaneously²⁰. A particularly straightforward method is to use the UCB (upper confidence bound) infill criterion²¹: Given a parameter λ , it uses the mean prediction of the surrogate model and adds λ times the model's uncertainty, thereby giving an optimistic bias to regions that have high uncertainty and therefore potential for improvement. By sampling multiple instances of the λ -parameter from an exponential distribution²², effectively making different tradeoffs between exploration and exploitation, one can generate different point propositions to be evaluated simultaneously. It is used by our method because, despite its simplicity, it is one of the best performing MBO parallelization methods²⁰."

Q19: *While differentiating between bacteria, phage and non-phage viral DNA can be argued to indeed be a taxonomic classification task, it works only on an extremely high level, while the reader could expect "taxonomic classification" to mean handling multiple hierarchical taxonomic units and multiple levels (down to individual species or even strains). It would be good to make this distinction more explicit.*

We followed your recommendation and changed the wording from "taxonomic classification" to "viral identification" or "viral classification" manuscript in multiple parts of the paper.

Q20: *The authors write that GMP (global max pooling) alone was shown to be worse GAP alone, but this usually depends on the task at hand. This may be the case for viral & taxonomic classification tasks, but could conceivably not hold for e.g. certain problems in regulatory genomics. Would it be possible to have GMP in GenomeNet-Architect as well and simply disable it when the user suspects it's not a good idea (as e.g. in the task selected by the authors)?*

We have updated the software to make different pooling possible. We have modified the manuscript as mentioned in our response to Q17.

Q21: *It is a bit surprising that Fiannaca-CNN performs so well, as one of the main strength of CNNs is translation invariance, while the k-mer occurrences in Fiannaca seem to be basically tabular data (the order of k-mer occurrence columns can usually be rearranged freely without loss of information). A filter (interpreted as a sliding window), operating on tabular data without the spatial structure that images or raw sequences have effectively has to deal with input having completely different 'meaning' depending on the filter's position. A comment on why this works nevertheless and what are the consequences of this working would be nice.*

We understand the point of the reviewer and we initially thought similarly about the architecture. We did not come up with this architecture, however, we believe a reason might be that, although the data is similar to the tabular as of type, the similarity between the features is more than in most use cases with tabular data. For example, in this model, the pre-processed input fed into a convolutional filter with a kernel size of five can iterate over the count numbers of "AAAAAA",

“AAAAAAC”, “AAAAAAT”, “AAAAAAG”, and “AAAAACA” one the one part of data, and “AAAATAA”, “AAAATAC”, “AAAATAT”, “AAAATAG”, and “AAAATCA” on another part of the data. Considering similarity of these data fragments to each other, there is potentially some amount of translated spatial structure to the pre-processed data fed into the model, although this pre-processed data is in fact tabular data. Nevertheless, it is important to note that we only include this method as a baseline and do not study why this method works rather well in-depth, as this study is out of the scope of this paper.

Q22: *I really appreciate how the authors aimed to analyze some trends in the optimized design choices (lines 301-308). It would be even more convincing if that analysis included some kind of a quantitative component, e.g. some measures of statistical significance supporting the claims. Otherwise, a comment explaining that the observed trends are not necessarily significant could help the reader critically assess the proposed interpretation.*

Instead of adding a quantitative component, we compared our statements with the optimized models on the new dataset/task (**Supplementary Table 1**) and observed that our findings are in line with the observed trends for the new dataset/task. We made the following modifications:

- Our optimization analysis shows that when compared to RNN models, architectures with GAP layers tend to use more convolutional layers (5 to 7 vs. 1 to 4) was changed to (5 to 7 vs. 1 to 5) as the detected architectures had 6 and 5 convolutional layers for the new task. Nevertheless, the observation on GAP models to have more convolutional layers than RNN ones is still valid.
- We added “for two datasets and three different sequence length values” to emphasize these trends are based on observations on both optimizations.
- We added “*It is important to note, however, that these trends are observations and may not universally apply to every dataset or task. Therefore, we recommend running GenomeNet-Architect on the specific dataset and task in question to tailor the model architecture for optimal performance in each unique scenario.*” as you requested.

The whole text is below:

*“In examining model architectures that perform well across two datasets and three different sequence lengths, we sought to identify common trends and patterns in their architecture designs. Our analysis reveals that architectures with GAP layers typically incorporate more convolutional layers (5 to 7, as opposed to 1 to 5 in RNN models) and more fully connected layers (1 or 2, vs. 0 or 1) (**Supplementary Table 1**). The preference for GAP layers is likely due to their function in aggregating information learned by convolutional layers through averaging over the sequence, instead of learning representations by optimizing its own weights. Compared to CNN-RNN models, CNN-GAP models mainly use fully connected layers to integrate long-range information. Furthermore, LSTM layers are consistently preferred to GRU layers. Our findings also indicate a general avoidance of multiple recurrent layers, while bidirectional RNN layers are preferred over unidirectional ones. In terms of training hyperparameters, the optimized learning rate is typically between 10^{-3} and 10^{-4} with the Adam³⁴ optimizer more*

commonly chosen over alternatives like Adagrad³⁵, Rmsprop, and SGD³⁶. It is important to note, however, that these trends are observations and may not universally apply to every dataset or task. Therefore, we recommend running GenomeNet-Architect on the specific dataset and task in question to tailor the model architecture for optimal performance in each unique scenario.”

Q23: *In Table 3, it is not fully clear if the numbers present the numbers of non-overlapping reads/fragments that can be theoretically generated (this seems to be the case), or the number of reads/fragments actually used to train the models (which could be higher, as mentioned, but potentially also lower). Making the distinction even more explicit would help.*

Table 2 indeed just describes the dataset and shows the number of non-overlapping subsequences that can be generated from the dataset to describe the size of the dataset for the reader. We updated the Datasets part to describe the content of Table 2 more clearly:

*“The number of FASTA files in the sets and the number of non-overlapping samples in sets of the viral classification task are listed in **Table 2**. Listed is the number of different non-overlapping sequences that could theoretically be extracted from the datasets, were they split into consecutive subsequences.”*

Q24: *Typos: In lines 191-192 “eukaryotic viruses (referred to as (bacterio-)phage) and prokaryotic viruses (referred to as viral non-phage DNA)” – should be the other way around.*

Thank you for spotting the typo. We have now modified as follows:

“When applied to the problem of viral identification task, distinguishing the origin of sequences from bacterial chromosomes from prokaryotic viruses (referred to as bacteriophage) and eukaryotic viruses (referred to as viral non-phage DNA), GenomeNet-Architect found architectures that outperformed other DL and non-DL methods we chose as baselines.”

Reviewer #2

Q1: *L29-41 (minor): To efficiently perform NAS ... two fully connected layers. I feel that this paragraph falls a little short from being a full review of methods that use DL in genomics, while also being very specific on a couple of niche subfields. I would suggest to add reference to a few general reviews of DL in genomics or specific subfields instead of talking about specific methods. Your method has the potential to be applied on many fields of genomics (and transcriptomics) so you are doing your manuscript a disservice with focusing too closely on one specific application.*

Thanks for the suggestion, we have now modified it as follows, clearly mentioning that there are numerous works in genomics to deep learning and added more references to several reviews of

DL as you suggested, such as these references: ¹¹⁻¹⁵. This section now reads:

“To efficiently perform NAS for ML tasks in genomics, it is essential to identify DL network architecture designs for genomic sequence analysis that are widely recognized in the literature. These designs often start with one or several convolutional layers, followed by a global pooling layer, and conclude with a series of fully connected layers²⁶⁻²⁹. Recurrent layers offer an alternative to convolutional or global pooling layers. Their ability to propagate information across sequences allows recurrent layers to effectively summarize data, comparable to pooling layers. While numerous works in genomics use RNN layers¹¹⁻¹⁵, one example is Seeker¹⁶, an RNN-based model which employs an LSTM layer for bacteriophage detection. Furthermore, integrating convolutional and recurrent layers, by stacking them sequentially, enhances model capability. For instance, the model developed by Wang et al.¹⁷ demonstrates this approach by placing an RNN on top of a convolutional layer, followed by two fully connected layers. A similar configuration is utilized in the DanQ model¹⁸, showcasing the effectiveness of combining recurrent and convolutional layers.”

Q2: L71-73 (minor): *No general purpose MBO-based NAS framework ... 2D image data instead*

I understand that this is the crux of the importance of this paper. I spent most of the introduction wondering “why not just use a generic NAS”. I would suggest to either put a mention of this in the abstract, and/or when you talk about different types of NNs and modalities of data (first paragraph) to drive home the point that genomic data is different from image or language data (and how). It will give more depth to the exploration of the question at hand.

This is a good point, we have now modified the following part of the abstract to include this information:

“[...] Here, we present GenomeNet-Architect, a neural architecture design framework that automatically optimizes deep learning models for genome sequence data. It optimizes the overall layout of the architecture, with a search space specifically designed for genomics. Additionally, it optimizes hyperparameters of individual layers and the model training procedure. [...]”

Q3: L96-98 (major): *“However, the resulting architecture can also be used for other tasks that are similar to the task for which it was optimized. It is therefore possible to perform a single optimization run to solve multiple genome sequence DL tasks.”*

I find this counter-intuitive in my experience with training neural networks for genomics tasks. It is uncommon that hyperparameters and architecture will be similar for conceptually similar tasks when training on different datasets. I would like to see at least some proof / demonstration of this sentence esp. since it is in the Results section.

We have added a new experiment on a related but different classification task (pathogenicity detection) which now demonstrates this point: our model optimized for viral classification task with 150 nt sequences also performs well on pathogenicity detection task on 250 nt sequences (Fig. 4). We have highlighted this finding in the following paragraph in the results section:

“Additionally, we adapted the models originally optimized for the viral classification (initially optimized for sequences of 150 nt, CNN-RNN-6h, and CNN-GAP-6h) by adjusting the input size to 250 nt to evaluate how well performance of an architecture optimized for one task transfers to a different task and conditions. These architectures are renamed to have “VC” (short for viral classification) as a suffix. The comparable performance of “VC” models to models detected by GenomeNet-Architect on this dataset “GAP-CNN” and “GAP-RNN” shows good transfer between related tasks on genome data (see Fig. 4).”

Besides this, we agree that having tasks that are too different may require different kinds of architectures.

Q4: L128 (minor): *You use interpolation for the hyperparameters between first and last layer. Even though I believe you based on practical experience that this works fine, it could be worthwhile to actually show that it does. You could possibly do one run on a dataset with interpolation and one with finding the HP for intermediate layers and report the differences in training time and HP prediction.*

We appreciate the suggestion regarding finding the hyperparameters of intermediate layers. However, designing a search space as suggested presents challenges. For instance, if optimizing the number of convolutional layers n_c from 0 to N , along with the kernel size of each layer (k_0, k_1, \dots, k_N), we face not only the issue of a high dimensional space and the "curse of dimensionality"¹⁹, but also the complexity of hyperparameter interdependencies. For example, when $n_c = 3$, parameters like k_4, k_5, \dots lose their effect, a scenario difficult for a black box optimizer to navigate efficiently. While alternative parameterization schemes could offer more control over individual layer hyperparameters and avoid such issues — for instance, defining k_0, k_{mid} , and k_{end} and applying quadratic interpolation — trying out all possible ways of doing this would be out of scope of our work. We do show that our current method already surpasses manually designed baseline models in performance.

Q5: L140 (minor): *I am looking at the architecture HP search space table, and I fail to see Dropout for the CNN layers. Isn't that a commonly used step in CNNs? Would it make sense in your proposed architectural search space?*

Dropout is not commonly used before or after convolutional layers. It is common to use batch normalization layers with convolutional layers as we did as a method for regularization. The ineffectiveness of dropout when used in convolutional layers was highlighted with the introduction of batch-normalization by Ioffe and Szegedy (2015)³⁰. This work states: “We have found that removing Dropout from BN-Inception allows the network to achieve higher validation accuracy. We conjecture that Batch Normalization provides similar regularization benefits as

Dropout, since the activations observed for a training example are affected by the random selection of examples in the same mini-batch.” Based on this understanding and the evidence presented, incorporating dropout into our convolutional layers setup within the proposed search space did not align with our optimization strategy.

Q6: L189 (no comment): I think that the showcase on taxonomic classification is nice, and it really drives home the message that this type of approach can actually improve predictive values.

We appreciate your positive evaluation.

Q7: Discussion (major): The whole discussion section is hyperfocused on the specific task that was used to showcase the method. I think that this does the manuscript a disservice. I would rather focus on many different potential uses of the method in various genomic and transcriptomic tasks. Since the method is generic, I believe that it has potential for many more applications.

We now compare our results against the models explored by our method on the pathogen/non-pathogen detection task. This is presented in the new subsection headed **“GenomeNet-Architect identifies models that outperform expert-designed baselines in the pathogenicity detection task”** in the results section. We used the dataset provided by the “DeePaC” study⁵ and compared our results with the benchmark carried out in this paper. See also the new Figure 4:

Performance on the pathogenicity detection task

Figure 4: Comparative analysis of misclassification rates in the pathogenicity detection task. Baseline models are represented in gray, while the red bars indicate models developed by GenomeNet-Architect. The data for the baseline results and the dataset itself are derived from the “DeePaC” study⁵. The graph shows individual model performance along with the enhanced accuracy archived through ensemble approaches. The graph underscores the superior performance of the GenomeNet-Architect models over different baselines.

Q8: *I would also like to see some discussion regarding how easy it is to expand the method to different types of neurons (e.g. GRU) and more complicated architectures (e.g. multiple input branches such as sequence, conservation) How easy is it to deploy and use. What type of equipment is needed to run the method at an acceptable timeframe.*

Regarding the use of GRU layers specifically, they are part of the search space presented in the paper (see Table 1); however, it turns out they are never selected as part of an "optimal" architecture. We did, however, add a sentence to the description of the search space highlighting the possibility of changing it:

"While the provided search space is our recommendation, our method also supports defining a custom search space, e.g. allowing more layers, or including GMP instead of GAP."

We have also changed the "Code Availability" section to give more information about deployment, in particular the possibility to use parallel GPUs and compute clusters:

"The code, available at <https://github.com/GenomeNet/Architect>, enables users to apply the optimization process across various datasets and tasks. It is based on our R library *deepG* (*deepg.de*) and can therefore be adapted to a variety of genomics tasks that are supported by it. It uses the TensorFlow backend and can be made to run in parallel on multi-GPU-machines and compute clusters through the *batchtools* R package."

The particular requirements for necessary resources are highly task-dependent, so we are not able to give a specific answer. As an orientation, we believe the description of the resource needs for our experiments in the "Hyperparameter optimization process" subsection are informative, e.g.:

"[...] For each task and sequence length value, the first $t = t_1$ (2 hours) optimization evaluated a total of 788 configurations, parallelized on 24 GPUs, and ran for 2.8 days (wall time). [...]"

Reviewer 3

Q1: *The author claimed that GenomeNet-Architect is a neural architecture design framework that researchers can use to automatically optimize learning models for genome sequence data. However, the framework is only applied to one task and thus this conclusion is not well-supported. Even for virus analysis, there are several different tasks, such as taxonomic classification and protein function annotation. Different tasks have different challenges, and the distributions of the data can vary a lot. If GenomeNet-Architect is a general framework, the author should show some benchmarks on at least two tasks. Considering that this can automatically optimize the architecture, this should be feasible.*

We have now extended our paper to another dataset. Please check our answer to **Reviewer 2 Q7**.

Q2: *The author should also compare it to other frameworks, such as DNABERT [1].*

[1] DNABERT: pre-trained Bidirectional Encoder Representations from Transformers model for DNA-language in genome

We are aware of DNABERT but think that it is not a suitable method to compare against:

The purpose of our paper is to come up with a framework that automatically designs suitable architectures for normal *supervised learning* on genomic data. DNABERT, on the other hand, is a *self-supervised* learning method: its purpose is to be pre-trained in a computationally intensive step on large amounts of unlabeled data, thereby learning intrinsic patterns in DNA sequences, that it then uses for data-efficient modeling in a later fine-tuning step. Therefore, these are models that solve different problems. Comparing models with pre-trained weights against ones that were not pre-trained (as the ones generated by GenomeNet-Architect) would not be a fair comparison.

One could consider comparing against a non-pretrained DNABERT, but the DNABERT paper itself shows that the DNABERT architecture does not work well without any pre-training (see Fig. 6.f in the paper: The "DNABERT-nopretrain" model vs. CNN, CNN+GRU or CNN+LSTM,

where “DNABERT-nopretrain” perform at least 10% worse than all these models): in this paper, the architecture is considerably outperformed by trivial CNN or CNN+LSTM models. We therefore do not find comparing against non-pretrained DNABERT worth the effort.

Comparing a pre-trained DNABERT against a hypothetical "pre-trained GenomeNet Architect model" would also not be of interest, since GenomeNet-Architect generates models that are optimized for normal supervised learning, not for efficient pretraining or fine-tuning. In this setting, we expect architectures that are specifically designed to be pre-trained to outperform architectures that were not.

Using GenomeNet-Architect to optimize a model specifically for efficient pre-training would also be an interesting research question. However, we consider this to be out of scope of this work, as this is a different machine learning paradigm and would also require a larger amount of computational resources.

Q3: *It is not clear how the authors generate the short/contig sequences. It is better to show contigs of different lengths (e.g. 2000, 5000 etc.).*

The "Architecture evaluation and benchmarks" subsection describes the process of training sequence generation:

“For the viral classification task, the training and validation samples are generated by randomly sampling FASTA genome files and splitting them into disjoint consecutive subsequences from a random starting point [...] Unlike during training and validation, the test set samples were not randomly generated by selecting random FASTA files. Instead, test samples were generated by iterating through all individual files, and using consecutive subsequences starting from the first position.”

We selected sequence lengths of 150 nt and 10,000 nt for our experiments because the former mirrors the typical length produced by short-read sequencing technologies, while the latter is representative of longer sequences such as assembled contigs. This choice allows us to assess the method's ability to handle both short sequences and longer ones, capturing long-distance relationships within the data.

To estimate how other length as suggested by the reviewer (e.g. 2000, 5000 etc.) would influence the results, we have conducted additional experiments to assess how sequence length influences the performance of our model, confirming its remarkable ability to generalize across a broad spectrum of sequence lengths.

Regarding intermediate sequence lengths (e.g., 2000, 5000 nt), our approach is versatile enough to optimize models for any given sequence length using GenomeNet-Architect, just as we have demonstrated for 150 nt and 10,000 nt sequences. The search space we defined is effective across these varying lengths without requiring manual adjustments for different sequence sizes.

Furthermore, models optimized for a specific length, such as 150 nt, can be adapted to classify longer sequences (e.g., 2000, 5000, 10000 nt). This adaptation can be achieved in several ways:

- *Sliding Window Approach*: Longer sequences can be segmented into shorter fragments matching the optimized model's input length (e.g., 150 nt). These fragments can then be processed individually, with their predictions aggregated to classify the original, longer sequence.
- *Retraining*: The model discovered for 150 nt can be trained from scratch for these desired longer sequences.
- *Fine-tuning*: The model trained on 150 nt sequences can then be fine-tuned on longer sequences.

We implemented all three aforementioned adaptation strategies on our most effective model, initially optimized for 150 nt sequences, to classify sequences of 10,000 nt. This approach provided us with a comprehensive set of results, including baseline comparisons for each adaptation method. For the *sliding window* approach, we employed a stride of 50 nt in our experiments. This technique involves creating overlapping subsequences to cover the entire length of the longer sequences being classified. For instance, to assess the model's performance on a 250 nt sequence, we generated three subsequences: from positions 1 to 150, 51 to 200, and 101 to 250. This method allowed us to systematically evaluate the model's ability to classify longer sequences without requiring additional training, by leveraging the predictive power developed for shorter sequences and applying it through a sliding window mechanism to cover the entirety of longer sequences.

In our study, we evaluated the performance of our best model, initially optimized for 150 nt sequences, on classifying 10,000 nt sequences using three distinct approaches: no additional training, training from scratch, and fine-tuning. The results from these methodologies yielded class-balanced accuracies of 97.7%, 98.4%, and 98.6%, respectively. These findings are particularly important as they demonstrate the robustness of our model across different sequence lengths. Notably, the training from scratch and fine-tuning methods resulted in performances that not only surpassed all but one of the expert-designed architectures but also achieved a comparable class-balanced accuracy of 98.6%, matching the performance of the Fiannaca baseline specifically for 10,000 nt sequences.

The experimental outcomes we have reported provide compelling evidence of our model's versatility and robustness across a broad spectrum of sequence lengths. Notably, our findings reveal that the model, initially trained solely on 150 nt sequences and not subjected to further training for longer sequences, delivers commendably across an array of sequence lengths.

The class-balanced accuracies recorded for this model across various sequence lengths are as follows: 79.8% for 150 nt, 84.1% for 250 nt, 89.8% for 500 nt, 93.2% for 1,000 nt, 95.3% for 2,000 nt, 96.8% for 5,000 nt, and 97.7% for 10,000 nt sequences. Remarkably, these accuracies were achieved without any additional training beyond the original model optimized for 150 nt sequences. This progression of performance across sequence lengths from 150 nt to 10,000 nt

not only attests to the model's capacity to generalize well beyond its initial training scope but also underscores its efficacy in handling considerably longer sequences with high accuracy.

Rebuttal Figure 1: Influence of sequence length to class balanced accuracy on the virus identification task

Lastly, we recognize the potential value of further evaluating GenomeNet-Architect through hyperparameter optimization for sequences of intermediate lengths. To this end, we have applied our method to a new dataset/task with a sequence length of 250 nt, as specified by the dataset's original creator. Our approach successfully identified models that surpassed the performance of existing baselines for this context. Moreover, we assessed the efficacy of models optimized for a different task/dataset (using hyperparameters optimized for a virus classification task) and for a different sequence length (150 nt instead of 250 nt) on this new dataset. Remarkably, these models also exceeded the performance of the baselines, as detailed in our response to **Reviewer 2, Q7**.

***Q4:** More detailed description of how the authors used the baseline methods should be provided. In the manuscript, the authors mentioned that they collected the data by themselves and split the data by time (before or after 2020). However, the mentioned baseline methods may be trained on different datasets based on different data collection and partition methods. The authors should retrain these baseline models on the same datasets for a fair comparison.*

We have ensured fairness in our comparisons by re-training all baseline models using the same datasets. We have implemented the architectures of these baseline methods and re-trained

them using the identical train/test/validation split that we employed for our own models. This approach guarantees that any performance comparisons are based on the same data and partitioning methods, thereby providing a more accurate and fair assessment of our optimized models against the established baselines.

See our response to your **Q5**, where we list the update we made to the manuscript to detail our experimental setup and further explain the training of our baselines.

Q5: *As described in Section Discussion, DeepMicrobes and CHEER are taxonomic classification tools. PPR-Meta is used to classify phages, chromosomes, and plasmids. Since their original design is not optimized for the virus identification task, the comparison may not be fair. At least re-training and adaption of these baseline models should be conducted. In addition, the author should include more virus identification tools for comparison if they focus on virus identification task.*

We apologize for not making this very clear. We have indeed conducted the experiments as suggested but may not have articulated this clearly in our initial manuscript. We now added the following clarification:

“For a fair comparison, we trained and validated all DL baseline models on the same dataset and dataset splits. Additionally, we standardized the configuration by adapting the output layer of each model and employing multi-class cross-entropy as the loss function, aligning with our models to facilitate three-class classification. This approach allows for a direct comparison of algorithmic improvements.”

Q6: *According to Table. 3. There are many more bacterial short/contig sequences in the datasets. How do the authors handle the data imbalance problem? The performance seems only for “balanced data”. Is the test data or the training data balanced? In real applications (such as metagenomic data), the label will not be balanced at all. How does this “balanced” performance translate to the real applications?*

The training and test data we are using is indeed imbalanced. The terminology may be a bit confusing here: "class-balanced accuracy" does not mean the "performance for balanced data"; instead it means that the accuracy for each class was weighted by an equal amount. This metric (class-balanced accuracy) is a frequently used metric to report the performance in such situations ^{10,16,31–33}.

We solve the imbalance problem by effectively oversampling during training: Although there is more bacterial data available, we sample the same number of contigs from each dataset, thereby potentially sampling any given piece of viral DNA more often than any piece of bacterial DNA.

We have updated the "Architecture evaluation and benchmarks" subsection to make this more clear:

"For the viral classification task, the training and validation samples are generated by randomly sampling FASTA genome files and splitting them into disjoint consecutive subsequences from a random starting point. A batch size that is a multiple of 3 (the number of target classes) is used, and each batch contains the same number of samples from each class. Since we work with datasets that have different quantities of data for each class, this effectively oversamples the minor classes compared to the largest class."

The test data is not balanced, which is specifically why we choose balanced accuracy as outcome measure. Unlike "normal" accuracy, it is independent of the class sizes in the test data, and unlike some other measures that are independent of class balance, like AUC-ROC, it straightforwardly generalizes to multiclass-classification.

Because the class imbalance relates to the dataset, we have added the following sentence to the "dataset" section:

"Because the size of the test set is imbalanced, we report class-balanced measures, i.e. measures calculated for each class individually and then averaged over all classes."

Q7: *More experiments should be established to demonstrate the robustness of the model. For example, how sequence similarity will affect the performance of the model. What is the performance of the real sequencing data?*

For details how sequence similarity affects the model performance, please see our additional experiments we describe **Reviewer 1, Q5**. Our experiments demonstrate that our model consistently surpasses the baseline across all intervals, particularly noting a considerable performance advantage in classifying test samples that bear less resemblance to the training set.

For a benchmark using simulated reads, please see response to **Reviewer 1, Q7**.

Q8: *Line 293: "performs better on taxonomic classification". There is no taxonomic classification result (e.g. family, genus, etc.) found in the manuscript. The task is more similar to virus identification.*

We followed your recommendation and changed the wording from "taxonomic classification" to "viral identification" or "viral classification" manuscript in multiple parts of the paper.

References

1. Wood, D. E. & Salzberg, S. L. Kraken: ultrafast metagenomic sequence classification using

- exact alignments. *Genome Biol.* **15**, 1–12 (2014).
2. Langmead, B. & Salzberg, S. L. Fast gapped-read alignment with Bowtie 2. *Nat. Methods* **9**, 357–359 (2012).
 3. Fiannaca, A. *et al.* Deep learning models for bacteria taxonomic classification of metagenomic data. *BMC Bioinformatics* **19**, 198 (2018).
 4. Bartoszewicz, J. M., Nasri, F., Nowicka, M. & Renard, B. Y. Detecting DNA of novel fungal pathogens using ResNets and a curated fungi-hosts data collection. *Bioinformatics* **38**, ii168–ii174 (2022).
 5. Bartoszewicz, J. M., Seidel, A., Rentzsch, R. & Renard, B. Y. DeePaC: predicting pathogenic potential of novel DNA with reverse-complement neural networks. *Bioinformatics* **36**, 81–89 (2019).
 6. Zhou, H., Shrikumar, A. & Kundaje, A. Towards a Better Understanding of Reverse-Complement Equivariance for Deep Learning Models in Genomics. in *Machine Learning in Computational Biology* 1–33 (PMLR, 2022).
 7. Katz, L. S. *et al.* Mashtree: a rapid comparison of whole genome sequence files. *J Open Source Softw* **4**, (2019).
 8. Henriksen, R. A., Zhao, L. & Korneliussen, T. S. NGSNGS: next-generation simulator for next-generation sequencing data. *Bioinformatics* **39**, (2023).
 9. Münch, P., Mreches, R., To, X. Y., Gündüz, H. A. & Moosbauer, J. A platform for deep learning on (meta) genomic sequences.
<https://www.researchsquare.com/article/rs-2527258/latest>.
 10. Gündüz, H. A. *et al.* A self-supervised deep learning method for data-efficient training in genomics. *Commun Biol* **6**, 928 (2023).
 11. Eraslan, G., Avsec, Ž., Gagneur, J. & Theis, F. J. Deep learning: new computational modelling techniques for genomics. *Nat. Rev. Genet.* **20**, 389–403 (2019).
 12. Koumakis, L. Deep learning models in genomics; are we there yet? *Comput. Struct.*

- Biotechnol. J.* **18**, 1466–1473 (2020).
13. Boža, V., Brejová, B. & Vinař, T. DeepNano: Deep recurrent neural networks for base calling in MinION nanopore reads. *PLoS One* **12**, e0178751 (2017).
 14. Cao, R. *et al.* ProLanGO: Protein Function Prediction Using Neural Machine Translation Based on a Recurrent Neural Network. *Molecules* **22**, (2017).
 15. Shen, X., Jiang, C., Wen, Y., Li, C. & Lu, Q. A Brief Review on Deep Learning Applications in Genomic Studies. *Frontiers in Systems Biology* **2**, (2022).
 16. Auslander, N., Gussow, A. B., Benler, S., Wolf, Y. I. & Koonin, E. V. Seeker: alignment-free identification of bacteriophage genomes by deep learning. *Nucleic Acids Res.* **48**, e121 (2020).
 17. Wang, R., Zang, T. & Wang, Y. Human mitochondrial genome compression using machine learning techniques. *Hum. Genomics* **13**, 49 (2019).
 18. Quang, D. & Xie, X. DanQ: a hybrid convolutional and recurrent deep neural network for quantifying the function of DNA sequences. *Nucleic Acids Res.* **44**, e107 (2016).
 19. Bischl, B. *et al.* Hyperparameter optimization: Foundations, algorithms, best practices, and open challenges. *Wiley Interdiscip. Rev. Data Min. Knowl. Discov.* **13**, e1484 (2023).
 20. Bischl, B., Wessing, S., Bauer, N., Friedrichs, K. & Weihs, C. MOI-MBO: Multiobjective Infill for Parallel Model-Based Optimization. in *Learning and Intelligent Optimization* 173–186 (Springer International Publishing, 2014).
 21. Srinivas, N., Krause, A., Kakade, S. M. & Seeger, M. Gaussian Process Optimization in the Bandit Setting: No Regret and Experimental Design. *arXiv [cs.LG]* (2009).
 22. Hutter, F., Hoos, H. H. & Leyton-Brown, K. Parallel Algorithm Configuration. in *Learning and Intelligent Optimization* 55–70 (Springer Berlin Heidelberg, 2012).
 23. Kingma, D. P. & Ba, J. Adam: A Method for Stochastic Optimization. *arXiv [cs.LG]* (2014).
 24. Stochastic Optimization. Adaptive Subgradient Methods for. <https://www.jmlr.org/papers/volume12/duchi11a/duchi11a.pdf> (2011).

25. Robbins, H. & Monro, S. A Stochastic Approximation Method. *Ann. Math. Stat.* **22**, 400–407 (1951).
26. Fang, Z. *et al.* PPR-Meta: a tool for identifying phages and plasmids from metagenomic fragments using deep learning. *Gigascience* **8**, (2019).
27. Tampuu, A., Bzhalava, Z., Dillner, J. & Vicente, R. ViraMiner: Deep learning on raw DNA sequences for identifying viral genomes in human samples. *PLoS One* **14**, e0222271 (2019).
28. Ren, J. *et al.* Identifying viruses from metagenomic data using deep learning. *Quant Biol* **8**, 64–77 (2020).
29. Shang, J. & Sun, Y. CHEER: HierarCHical taxonomic classification for viral mEtagEnomic data via deep leaRning. *Methods* **189**, 95–103 (2021).
30. Ioffe, S. & Szegedy, C. Batch Normalization: Accelerating Deep Network Training by Reducing Internal Covariate Shift. in *International Conference on Machine Learning* 448–456 (PMLR, 2015).
31. Bartoszewicz, J. M., Seidel, A. & Renard, B. Y. Interpretable detection of novel human viruses from genome sequencing data. *NAR Genom Bioinform* **3**, lqab004 (2021).
32. Narayan, N. R. *et al.* Correction to: Piphillin predicts metagenomic composition and dynamics from DADA2- corrected 16S rDNA sequences. *BMC Genomics* **21**, 105 (2020).
33. Lewis, J. E. & Kemp, M. L. Integration of machine learning and genome-scale metabolic modeling identifies multi-omics biomarkers for radiation resistance. *Nat. Commun.* **12**, 2700 (2021).

REVIEWERS' COMMENTS:

Reviewer #1 (Remarks to the Author):

I would like to thank the authors for meticulously addressing all the comments and questions. I especially appreciate the discussion regarding why Fiannaca works in practice. While in Q4, my goal was to suggest another architecture only (rather than an additional benchmarking task/dataset as well), I find new results are very interesting and valuable. As for alternative tasks/data, the goal was rather to suggest something related to eukaryotic/regulatory genomics, or actually anything of interest for the authors and/or readers. My most sincere apologies for the misunderstanding and any and all inconvenience caused by having to run more experiments. In the end, the authors have convincingly shown that their approach works for more than one task (which was a separate issue raised by other reviewers as well), so I hope this was not in vain.

The manuscript seems ready for publication - thank you for the great work!

The only very minor feedback: while the authors ended up running experiments on three sequence lengths (150, 250, 10k), some parts of the manuscript still refer only to the original two. It could be easier for the readers to correct that - but it can easily be done in production.

Reviewer #2 (Remarks to the Author):

Given the replies of the authors, all my concerns have been addressed.

Reviewer #3 (Remarks to the Author):

The authors have answered most of my questions and revised the manuscript accordingly. I appreciate their efforts. But there are still a couple remaining questions. Please see below.

Major:

1. The author may have misunderstood the usage of DNABERT. Self-supervised training is a strategy for deep learning models to find better initial points, which can help the model converge better in downstream supervision tasks. Like the authors claim that they come up with a framework that automatically designs suitable architectures for normal supervised learning on genomic data, DNABERT has the same capabilities by fine-tuning the model to specific tasks, such as virus identification and pathogenicity detection tasks. Thus, the authors can fine-tuning the DNABERT on their data with the same training process to show a more comprehensive benchmark performance.

Minor:

In line 199: "distinguishing the origin of sequences from bacterial chromosomes from prokaryotic viruses (referred to as bacteriophage)". Hard to follow.

Responses to Reviewer Feedback

Reviewer #1

I would like to thank the authors for meticulously addressing all the comments and questions. I especially appreciate the discussion regarding why Fiannaca works in practice. While in Q4, my goal was to suggest another architecture only (rather than an additional benchmarking task/dataset as well), I find new results are very interesting and valuable. As for alternative tasks/data, the goal was rather to suggest something related to eukaryotic/regulatory genomics, or actually anything of interest for the authors and/or readers. My most sincere apologies for the misunderstanding and any and all inconvenience caused by having to run more experiments. In the end, the authors have convincingly shown that their approach works for more than one task (which was a separate issue raised by other reviewers as well), so I hope this was not in vain.

The manuscript seems ready for publication - thank you for the great work!

We appreciate your positive evaluation.

Q1: The only very minor feedback: while the authors ended up running experiments on three sequence lengths (150, 250, 10k), some parts of the manuscript still refer only to the original two. It could be easier for the readers to correct that - but it can easily be done in production.

We have made minor corrections throughout the manuscript to consistently refer to all three sequence lengths (150nt, 250 nt and 10k nt) where appropriate. However, in some instances, we specifically discuss the viral classification task, which only involved 150 nt and 10k nt sequence length. In these cases, no corrections were necessary.

Reviewer #2

Given the replies of the authors, all my concerns have been addressed.

We are delighted to hear that all your concerns have been addressed satisfactorily. Thank you for your positive evaluation of our revised manuscript.

Reviewer #3

The authors have answered most of my questions and revised the manuscript accordingly. I appreciate their efforts. But there are still a couple remaining questions. Please see below.

Major:

Q1: *The author may have misunderstood the usage of DNABERT. Self-supervised training is a strategy for deep learning models to find better initial points, which can help the model converge better in downstream supervision tasks. Like the authors claim that they come up with a framework that automatically designs suitable architectures for normal supervised learning on genomic data, DNABERT has the same capabilities by fine-tuning the model to specific tasks, such as virus identification and pathogenicity detection tasks. Thus, the authors can fine-tuning the DNABERT on their data with the same training process to show a more comprehensive benchmark performance.*

We appreciate your recognition of our efforts in addressing most of your questions and revising the manuscript accordingly. We have carefully considered your remaining concerns and have made the following changes:

Regarding your suggestion to fine-tune DNABERT for a more comprehensive benchmark, we have followed your advice and included DNABERT¹ (6-mer model) as an additional baseline for the pathogenicity detection task. Our optimized models outperform the fine-tuned DNABERT baseline, further demonstrating the effectiveness of our approach. We have updated our manuscript, figures and figures descriptions to reflect this additions

“[...] We also fine-tuned the pre-trained DNABERT¹ (6-mer model), using the suggested hyperparameter settings given for fine-tuning on the method's GitHub page. We added it as an additional baseline for this task to make our benchmark more comprehensive.”

We also updated the relevant figure and the figure description accordingly, as below:

Performance on the pathogenicity detection task

Figure 4: Comparative analysis of misclassification rates in the pathogenicity detection task. The baseline models are shown in gray, while the red bars indicate the models developed by GenomeNet-Architect. The data for the dataset itself and the baseline results, with the exception of DNABERT¹, were derived from the DeePaC study². In addition, the pre-trained DNABERT¹ model is fine-tuned on this task to make the benchmark more comprehensive. The graph shows individual model performance along with the improved performance archived by the ensemble approaches and highlights the superior performance of the GenomeNet-Architect models over various baselines.

Minor:

Q2: In line 199: “distinguishing the origin of sequences from bacterial chromosomes from prokaryotic viruses (referred to as bacteriophage) “. Hard to follow.

To address your concern about the clarity of the sentence, we have rephrased it as follows:

“GenomeNet-Architect demonstrated superior performance on the virus classification task compared to other deep learning (DL) and non-deep learning methods that we selected as baselines, effectively distinguishing between sequences originating from bacterial chromosomes, prokaryotic viruses (referred to as bacteriophages) and eukaryotic viruses (referred to as viral non-phage DNA).”

We hope that these changes adequately address your remaining concerns and enhance the overall quality of our manuscript.

References

1. Ji, Y., Zhou, Z., Liu, H. & Davuluri, R. V. DNABERT: pre-trained Bidirectional Encoder Representations from Transformers model for DNA-language in genome. *Bioinformatics* (2021) doi:10.1093/bioinformatics/btab083.
2. Bartoszewicz, J. M., Seidel, A., Rentsch, R. & Renard, B. Y. DeePaC: predicting pathogenic potential of novel DNA with reverse-complement neural networks. *Bioinformatics* **36**, 81–89 (2019).